# The Cost of Commitment in Option-Based Hierarchical RL

**Randy Lefebvre** [1 2 3]   **Audrey Durand** [1 2 3 4]

## Abstract

Empirically, option-based hierarchical reinforcement (HRL) learning often produces longer and more diverse options when a deliberation cost is charged at option boundaries. However, when options are executed for many steps under an approximate dynamics model, small model errors compound along the option, degrading the quality of the resulting plan. In this work, we introduce the *commitment loss* to formalize the tradeoff between deliberation cost and model error as a function of option duration. We characterize how optimal termination probabilities vary with this tradeoff under two model-error mechanisms. First, the model is learned from finite data via maximum-likelihood estimation, producing statistical error that interacts with option duration. Second, we consider an input-driven setting where an exogenous input is only observed at option boundaries and evolves unobserved between them, creating a drift-induced mismatch between planned and realized dynamics. In both cases, we solve for the optimal termination behavior as a function of deliberation cost and the error scale, clarifying the behavior of some popular HRL algorithms that approach the deliberation cost as a heuristic.

Reinforcement learning (RL) is increasingly used to tackle complex control problems, from robotics (Joshi et al., 2020; Schoettler et al., 2020) to large language model (LLM) agents acting in open-ended environments (Wang et al., 2023). Despite this progress, the poor sample efficiency of most RL methods and the difficulties arising from long-horizon credit assignment still limit the deployment of RL to real world scenarios (Dulac-Arnold et al., 2019). Hierarchical reinforcement learning (HRL) addresses these

challenges by introducing temporal abstractions in the form of skills, options, or macro actions (Sutton et al., 1999; Nachum et al., 2018; Wang et al., 2025). Instead of selecting primitive actions at every time step, a high level policy chooses among temporally extended options that commit to a course of action for multiple steps (Hutsebaut-Buysse et al., 2022). By reasoning at a coarser timescale, HRL concentrates the learning signal on a smaller number of high-level decisions, which can significantly improve sample efficiency and facilitate credit assignment in long and sparse reward tasks (Biedenkapp et al., 2021).

Unfortunately, discovering such options remains difficult. Although most HRL formulations assume that the cost of switching between options (or recomputing the high-level policy) is negligible (Evans & Şimşek, 2023; Nachum et al., 2018), it has been shown that options tend to collapse to individual actions under this assumption, terminating at every time step (Harb et al., 2018). To prevent this degeneracy, Harb et al. (2018) introduced a deliberation cost that is incurred whenever the agent switches to a new option. This matches many real world settings: in robotics, running Simultaneous Localization and Mapping (SLAM) to access the current location to make a short term plan is costly (Yanık & Ilgın, 2023); in LLM-based agents, each deliberation phase (e.g., chain-of-thought planning with tool use) introduces non-trivial latency and monetary cost, in addition to the expense of requesting human feedback (Nayab et al., 2024). In such cases, it is natural to commit to an option for more than one time step.

Yet committing for too long has its own cost. When options are executed for many steps under an approximate dynamics model, small model errors compound along the option, degrading the quality of the resulting plan. Even with a perfect model, exogenous inputs and context information may drift over time, so a policy that was optimal at option initiation can become increasingly misaligned with the current environment as the option progresses. In standard, non-hierarchical RL, the impact of the planning horizon on the value-function estimation error and variance has been extensively analyzed, most often through the role of the discount factor (Petrik & Scherrer, 2008; Jiang et al., 2015; Lefebvre & Durand, 2025). In contrast, the analogous question at the level of option durations, that is, how long it is safe to commit to an option before the model error or drift

---

[1]Département d'informatique et de génie logiciel, Université Laval [2]Mila – Institut Québécois d'IA [3]Institut Intelligence et Données (IID), Université Laval [4]Canada CIFAR AI Chair. Correspondence to: Randy Lefebvre <randy.lefebvre.1@ulaval.ca>, Audrey Durand <audrey.durand@ift.ulaval.ca>.

*Proceedings of the 43$^{rd}$ International Conference on Machine Learning*, Seoul, South Korea. PMLR 306, 2026. Copyright 2026 by the author(s).

dominates the re-planning cost, has received comparatively little attention. The closest work we are aware of is Harutyunyan et al. (2019), who study the impact of discounting schemes in options w.r.t. option duration.

**Contributions** Our work provides, to the best of our knowledge, the first theoretical framework to formalize the tradeoff between deliberation cost and model error as a function of option duration. To achieve this, we introduce the *commitment loss*, which measures the loss incurred when planning on an approximate model with a restricted option set (shorter options) compared to an unconstrained baseline evaluated on the ground-truth dynamics. We decompose the commitment loss into a *deliberation cost*, which grows as options are forced to terminate more frequently, and the *model variance*, which captures the compounding effect of model error along long-running options; this yields a general bound on the deliberation cost. We then conduct a theoretical analysis of the model variance resulting from i) an approximate model and ii) information drift.

In the first setting, the approximate model is learned from finite data via maximum likelihood estimation. We provide an upper bound on the model variance, leading to a PAC-style upper bound on the commitment loss. This result highlights the impact of the option termination rate on the tradeoff between deliberation cost and statistical error given the sample size. In the second setting, we consider an input-driven environment and analyze the variance under a frozen-input approximation model, where inputs are treated as constant within each option. This is akin to assuming that inputs are only available at option decisions. In this regime, the deliberation cost turns into an *information cost*, yielding a tradeoff between the cost for acquiring information and the loss from partial observability. In both settings, we solve the resulting upper bounds on the commitment loss to obtain explicit recommendations for option duration, and we empirically validate these predictions in controlled experiments. Our implementation is available on github.

## 1. Background

Consider a finite discounted MDP $M = (\mathcal{S}, \mathcal{A}, P, R, \gamma)$, where $\mathcal{S}$ is the state space, $\mathcal{A}$ is the action space, $P : \mathcal{S} \times \mathcal{A} \to \Delta(\mathcal{S})$ is the transition kernel, $R : \mathcal{S} \times \mathcal{A} \to [0, R_{\max}]$ is a bounded reward function, and $\gamma \in [0, 1)$ is the discount factor. At each time step $t \in \mathbb{N}_0$, the agent observes the current state $S_t$ and chooses an action $A_t = \pi(S_t)$ according to a deterministic policy $\pi : \mathcal{S} \to \mathcal{A}$. The environment then transitions into the next state $S_{t+1} \sim P(\cdot \mid S_t, A_t)$ and the agent receives the reward $R_{t+1} = R(S_t, A_t)$. The value of a state $s \in \mathcal{S}$ under policy $\pi$ in MDP $M$ is the expected sum of discounted rewards:

$$V_M^\pi(s) := \mathbb{E}_\pi \left[ \sum_{k=0}^\infty \gamma^k R_{t+k+1} \,\middle|\, S_t = s \right].$$

An optimal policy $\pi_M^\star$ satisfies $V_M^{\pi_M^\star}(s) \geqslant V_M^\pi(s)$ for all states $s \in \mathcal{S}$ and all policies $\pi$ on MDP $M$.

**Options** The options framework introduces the concept of temporally extended actions (Sutton et al., 1999). Formally, an option $o$ is defined by the set of states $\mathcal{I}_o \subseteq \mathcal{S}$ where the option can be initiated, a (deterministic) intra-option policy $\pi_o : \mathcal{S} \to \mathcal{A}$, and a termination function $\beta_o : \mathcal{S} \to [0, 1]$ indicating the probability of terminating the option in each state. Let $\Pi$ denote the finite set of intra-option policies, and let $\mathcal{O}$ be the class of options. Similar to prior work (Sutton et al., 1999; Bacon et al., 2017), we assume that options can be initiated in every state, i.e., $\mathcal{I}_o = \mathcal{S}, \ \forall o \in \mathcal{O}$. Under this assumption, options are fully specified by $(\pi_o, \beta_o)$.

**Semi-Markov Decision Process (SMDP)** Options are typically formalized under the SMDP setting (Howard, 1963; Puterman, 1994), which augments the MDP with options. In such SMDP, the agent chooses options according to a (deterministic) meta-policy $\mu : \mathcal{S} \to \mathcal{O}$. Following Harb et al. (2018), we consider that every decision made by the meta-policy has a cost, which is defined by the deliberation cost function $C : \mathcal{S} \times \mathcal{O} \to [0, C_{\max}]$. Let $\{T_0, T_1, T_2, \dots\}$ be the random sequence of option initiation times. At time $T_n$, the agent observes the state $S_{T_n}$, selects the option $O_n = \mu(S_{T_n})$, incurs a deliberation cost $C(S_{T_n}, O_n)$, and then executes primitive actions according to the intra-option policy $\pi_{O_n}$ until termination of the option. During the execution of an option, rewards and transitions are generated as in the underlying MDP and no additional deliberation cost is incurred. We model deliberation cost as an instantaneous penalty at option initiation:

$$C_t := \begin{cases} C\left(S_t, \mu(S_t)\right), & \text{if } t = T_n \text{ for some } n, \\ 0, & \text{otherwise.} \end{cases}$$

Given a base MDP $M$ and an option class $\mathcal{O}$, the value of a state $s \in \mathcal{S}$ under meta-policy $\mu$ is given by

$$V_{M,\mathcal{O}}^\mu(s) = \mathbb{E}_\mu \left[ \sum_{k=0}^\infty \gamma^k \left( R_{t+k+1} - C_{t+k} \right) \,\middle|\, S_t = s \right].$$

We keep the notation $V_{M,\mathcal{O}}^\mu$ to emphasize the distinction between the environment on which the policy is optimized and the one on which it is evaluated. An optimal meta-policy $\mu_{M,\mathcal{O}}^\star$ satisfies $V_{M,\mathcal{O}}^{\mu_{M,\mathcal{O}}^\star}(s) \geqslant V_{M,\mathcal{O}}^\mu(s)$ for all $s \in \mathcal{S}$ and all meta-policies $\mu$. In this work, our various environments will vary in their underlying dynamics $M$ and option class $\mathcal{O}$, but the cost function $C$ remains the same for all.

## 2. Commitment Loss in Option-Based SMDPs

Long-term options are beneficial in the sense that they allow the agent to pay less deliberation cost. However, model error may compound, negatively impacting returns. We formulate the resulting tradeoff, which we call the commitment loss, under specific assumptions on the structure of the option-based SMDP. We then provide an upper bound on the deliberation cost as a function of option durations.

### 2.1. Structure and Notation

To enable our analysis, we consider SMDPs with two desirable properties: 1) option durations that are independent of the intra-option policies; and 2) a finite option class parameterizable by option durations.

**Assumption 2.1** (State-independent termination). Let $B \subset (0, 1]$ be a finite set. For every option $o \in \mathcal{O}$, the termination function is constant over states and satisfies

$$\beta_o(s) \equiv \beta(o) \in B \qquad \forall s \in \mathcal{S}.$$

This assumption implies that once an option $o \in \mathcal{O}$ is initiated, each step terminates the option independently with probability $\beta(o)$. Options constructed from action repetitions (Biedenkapp et al., 2021) typically satisfy this assumption. More generally, state-independent terminations can be found in any setting where the commitment length is a design choice decoupled from the skill itself. Although quite restrictive, removing this assumption leads to a tight coupling between the option's duration, the policy and the state structure. In this work, we focus on a domain agnostic theoretical understanding of how option duration affects deliberation costs and compounding errors. We leave the exploration of domain specific bounds for future work. Given state-independent terminations, it is natural to talk about the *duration*[1] of an option, which is on the order of $\frac{1}{\beta(o)}$. Intuitively, if $\beta(o) < \beta(o')$ for $o, o' \in \mathcal{O}$, then option $o$ has a longer expected duration than option $o'$.

**Definition 2.2** (Duration-specified options). Under Assumption 2.1 and $\mathcal{I}_o = \mathcal{S}$, we consider the option class

$$\mathcal{O} := \Pi \times B,$$

where each option is specified by an intra-option policy $\pi \in \Pi$ and a termination probability $\beta \in B$.

Given duration-specified options, the intra-option policies $\Pi$ are fixed and reused at all admissible durations induced by the termination probabilities in $B$. We can then use $B$ to parametrize our option class.

---

[1]We will sometimes use $\beta$ and "duration" interchangeably, with smaller $\beta$ corresponding to longer option duration.

**Limiting option durations** For any threshold $\beta' \in B$, define the set of lower-bounded termination probabilities (upper-bounded durations) and the associated subclass of options with limited durations as

$$B_{\geqslant \beta'} := \{\beta \in B : \beta \geqslant \beta'\} \quad \text{and} \quad \mathcal{O}_{\geqslant \beta'} := \Pi \times B_{\geqslant \beta'}.$$

Increasing $\beta'$ thus shrinks the option class and to options with shorter commitments, with the extreme $\beta' = 1$ collapsing to only options of duration one.

**Note 2.3.** *The behavior of any option $(\pi, \beta) \in \mathcal{O}$ can be retrieved by sequentially re-selecting a shorter version of the same option, that is $(\pi, \beta')$ for any $\beta' > \beta$. Therefore, all behaviors achievable by optimizing a meta-policy on $\mathcal{O}$ can also be obtained by optimizing a meta-policy on $\mathcal{O}_{\geqslant \beta'}$.*

### 2.2. Commitment Loss

Let $\beta_{\min} := \min B$ denote the termination probability defining the longest available duration. Note that $\mathcal{O}_{\geqslant \beta_{\min}}$ corresponds to the complete option class $\mathcal{O}$. Let $\beta_{\mathrm{eval}} \in B$ with $\beta_{\mathrm{eval}} \geqslant \beta_{\min}$ denote the *evaluation* threshold, scoping the option class. For brevity, we write $\mathcal{O}_{\mathrm{eval}} := \mathcal{O}_{\geqslant \beta_{\mathrm{eval}}}$.

**Definition 2.4** (Commitment loss). Fix an MDP $M$, an option class $\mathcal{O}$ satisfying Definition 2.2, and a deliberation cost function $C$. The commitment loss captures the impact of learning a meta-policy on a subclass class of options with limited durations, $\mathcal{O}_{\mathrm{eval}}$, and an imperfect model $\widehat{M} \approx M$:

$$\left\| V_{M,\mathcal{O}}^{\mu_{M,\mathcal{O}}^{\star}} - V_{M,\mathcal{O}}^{\mu_{\widehat{M},\mathcal{O}_{\mathrm{eval}}}^{\star}} \right\|_{\infty}. \tag{1}$$

To provide insight on the optimal option duration, we decompose the commitment loss into a term scaling with the deliberation costs and another with model variance:

$$\left\| V_{M,\mathcal{O}}^{\mu_{M,\mathcal{O}}^{\star}} - V_{M,\mathcal{O}}^{\mu_{\widehat{M},\mathcal{O}_{\mathrm{eval}}}^{\star}} \right\|_{\infty} \leqslant \tag{2}$$

$$\underbrace{\left\| V_{M,\mathcal{O}}^{\mu_{M,\mathcal{O}}^{\star}} - V_{M,\mathcal{O}}^{\mu_{M,\mathcal{O}_{\mathrm{eval}}}^{\star}} \right\|_{\infty}}_{\text{deliberation cost}} + \underbrace{\left\| V_{M,\mathcal{O}_{\mathrm{eval}}}^{\mu_{M,\mathcal{O}_{\mathrm{eval}}}^{\star}} - V_{M,\mathcal{O}_{\mathrm{eval}}}^{\mu_{\widehat{M},\mathcal{O}_{\mathrm{eval}}}^{\star}} \right\|_{\infty}}_{\text{variance}}.$$

We get the result by a simple triangle inequality (see Appendix A). The deliberation cost captures the loss from using shorter options, resulting into more decisions by the meta-policy. The variance broadly captures the compounding error arising from long-term planning on a model that poorly mirrors ground-truth. This decomposition cleanly highlights a tradeoff offered between options with long duration (reducing deliberation costs) and shorter options (reducing variance). We provide the following upper bound on the decision cost.

**Lemma 2.5** (Deliberation cost). *Fix an MDP $M$, an option class $\mathcal{O}$ as in Definition 2.2, and a deliberation cost function $C : \mathcal{S} \times \mathcal{O} \to [0, C_{\max}]$. The deliberation cost incurred*

*by optimizing the meta-policy on a subclass of options with limited durations, $\mathcal{O}_{\text{eval}}$, is bounded by*

$$\left\| V_{M,\mathcal{O}}^{\mu_{M,\mathcal{O}}^{\star}} - V_{M,\mathcal{O}}^{\mu_{M,\mathcal{O}_{\text{eval}}}^{\star}} \right\|_{\infty} \leqslant \frac{\gamma\, C_{\max}}{1-\gamma}\left(\beta_{\text{eval}} - \beta_{\min}\right).$$

The proof relies Definition 2.2 and Note 2.3, observing that the optimal behavior on $\mathcal{O}$ can be achieved using a larger combinations of options from $\mathcal{O}_{\text{eval}}$ allows to cancel all primitive rewards and expose the differences in deliberation cost, under a simple coupling argument (see Appendix B). Lemma 2.5 shows that the deliberation cost grows linearly with the termination probability threshold $\beta_{\text{eval}}$: restricting the agent to shorter options using higher termination probability forces the agent to pay deliberation costs more frequently. When $C_{\max}$ is small, this penalty is mild and shorter options may be acceptable. When $C_{\max}$ is large, the bound favours longer options. The strong dependence on the discount factor $\gamma$ also resonates with prior work linking option durations and discounting (Harutyunyan et al., 2019).

## 3. Cost-Variance Tradeoff from Model Error

We now instantiate an approximate model $\widehat{M}$ of the true underlying MDP $M = (\mathcal{S}, \mathcal{A}, P, R, \gamma)$. We provide an upper bound on the variance (Equation 2) given that planning is performed with $\widehat{M}$ over a subclass of options with limited durations, $\mathcal{O}_{\text{eval}} \subseteq \mathcal{O}$, induced by $B$ via the threshold $\beta_{\text{eval}}$.

**Approximate model from finite data**  We consider a standard model-based setting where $\widehat{M}$ is obtained from $M$ by estimating the transition kernel from finite data. For each $(s, a) \in \mathcal{S} \times \mathcal{A}$ we observe $n$ i.i.d. samples of the next state and construct the maximum-likelihood estimate $\widehat{P}(\cdot \mid s, a)$. The approximate MDP is thus $\widehat{M} = (\mathcal{S}, \mathcal{A}, \widehat{P}, R, \gamma)$ and differs from $M$ only in $P$. Under this approximate model, we obtain the following PAC-bound on the variance.

**Lemma 3.1** (Variance). *Fix an MDP $M = (\mathcal{S}, \mathcal{A}, P, R, \gamma)$ and an option class $\mathcal{O}_{\text{eval}}$ as in Definition 2.2. Suppose that the approximate model $\widehat{M} \approx M$ is obtained by maximum-likelihood estimation of the transition kernel from $n$ samples per state–action pair. Then, with probability at least $1 - \delta$, the variance (Equation 2) satisfies*

$$\left\| V_{M,\mathcal{O}_{\text{eval}}}^{\mu_{M,\mathcal{O}_{\text{eval}}}^{\star}} - V_{M,\mathcal{O}_{\text{eval}}}^{\mu_{\widehat{M},\mathcal{O}_{\text{eval}}}^{\star}} \right\|_{\infty}$$
$$\leqslant \frac{2\gamma R_{\max}}{(1-\gamma)^2 \beta_{\text{eval}}} \sqrt{\frac{2}{n}\left(\log(2^{|\mathcal{S}|} - 2) + \log\frac{|\mathcal{S}||\mathcal{A}|}{\delta}\right)}.$$

The proof uses perturbation bounds for Markov chains to lift a one-step deviation between $P$ and $\widehat{P}$ to a deviation between the corresponding option-level transition kernels (see Appendix C). Smaller termination probabilities (longer

options) discount the per-step deviations less aggressively, leading to the $1/\beta_{\text{eval}}$ dependence in the bound. For a fixed sample size $n$, the bound favors shorter options (larger $\beta_{\text{eval}}$) and, as in Jiang et al. (2015), favors a smaller effective horizon (i.e., smaller $\gamma$). This must however be balanced against the deliberation costs captured by Lemma 2.5.

**Cost–variance tradeoff**  Combining the deliberation-cost bound (Lemma 2.5) with the variance bound (Lemma 3.1) via Equation 2 yields a PAC-bound on the commitment loss.

**Theorem 3.2** (Commitment loss from model error). *Fix an MDP $M = (\mathcal{S}, \mathcal{A}, P, R, \gamma)$ and an option class $\mathcal{O}$ as in Definition 2.2, with $\mathcal{O}_{\text{eval}} \subseteq \mathcal{O}$ induced by the threshold $\beta_{\text{eval}} \in B$. Suppose that the approximate model $\widehat{M} \approx M$ is obtained by maximum-likelihood estimation of the transition kernel from $n$ samples per state–action pair. Then, with probability at least $1 - \delta$, the commitment loss satisfies*

$$\left\| V_{M,\mathcal{O}}^{\mu_{M,\mathcal{O}}^{\star}} - V_{M,\mathcal{O}}^{\mu_{\widehat{M},\mathcal{O}_{\text{eval}}}^{\star}} \right\|_{\infty}$$
$$\leqslant \frac{\gamma C_{\max}}{1-\gamma}\left(\beta_{\text{eval}} - \beta_{\min}\right)$$
$$+ \frac{2\gamma R_{\max}}{(1-\gamma)^2 \beta_{\text{eval}}} \sqrt{\frac{2}{n}\left(\log(2^{|\mathcal{S}|} - 2) + \log\frac{|\mathcal{S}||\mathcal{A}|}{\delta}\right)}.$$

The result follows by substituting the corresponding bounds into Equation 2 (see Appendix D). The first term grows linearly with the termination probability threshold $\beta_{\text{eval}}$, capturing the deliberation cost induced by enforcing shorter options. The second term decreases at rate $1/\beta_{\text{eval}}$, reflecting how increasing termination frequency reduces the accumulation of model error along an option.

**Corollary 3.3** (Bound-optimizing termination rate). *Under the assumptions of Theorem 3.2, and for fixed $(\gamma, |\mathcal{S}|, |\mathcal{A}|, \delta)$, assume that the minimizer of the bound in Theorem 3.2 lies in the interior $(\beta_{\min}, 1)$. Then,*

$$\beta_{\text{eval}}^{\star} = \Theta\left(\sqrt{\frac{R_{\max}}{C_{\max}}}\, n^{-1/4}\right),$$

*where the hidden constants depend only on $(\gamma, |\mathcal{S}|, |\mathcal{A}|, \delta)$. When the unconstrained minimizer falls outside $[\beta_{\min}, 1]$, the optimal $\beta_{\text{eval}}^{\star}$ simply saturates at one of the endpoints, that is $\beta_{\min}$ or 1.*

The result follows by differentiating the bound in Theorem 3.2 with respect to $\beta_{\text{eval}}$ (see Appendix E). Corollary 3.3 makes the dependence of the bound-optimizing termination rate on the problem parameters explicit. When deliberation costs are large compared to the reward scale (large $C_{\max}$ for fixed $R_{\max}$), the factor $\sqrt{R_{\max}/C_{\max}}$ is small and the bound is minimized by a small $\beta_{\text{eval}}^{\star}$, i.e., longer options that incur deliberation costs less frequently.

Conversely, when deliberation is cheap relative to rewards (small $C_{\max}$), $\beta^{\star}_{\text{eval}}$ moves closer to 1, favoring shorter options and more frequent replanning. The dependence on the sample size is also explicit: $\beta^{\star}_{\text{eval}}$ shrinks only at a slow $n^{-1/4}$ rate, so a substantial increase in sample size $n$ is required for the bound to strongly favour long commitments.

# 4. Cost-Variance Tradeoff under Information Drift in Input-Driven SMDPs

Over the years, various extensions of the classical MDP have been introduced to capture important dynamics arising in real-world problems. In the Input-Driven MDP (IDMDP) setting (Mao et al., 2018), state variables controlled by the agent (states) are modeled independently of those that the agent does not control (inputs). Inputs are typically rich observations generated by an external stochastic processes that are difficult to model accurately, but that provide contextual information to the agent for making decisions. In this section, we extend the IDMDP setting to the options framework. We then focus on a situation where the agent relies on an approximate model of the true IDMDP and provide an upper bound on the resulting variance (Equation 2).

## 4.1. Input-driven SMDP (IDSMDP)

Recall that an option-based SMDP (Section 1) relies on a base MDP augmented with options. Similarly, we introduce the IDSMDP, which relies on a base IDMDP $M = (\mathcal{S}, \mathcal{Z}, \mathcal{A}, P_s, P_z, R, \gamma)$, where $\mathcal{S}$ is the state space, $\mathcal{Z}$ is the input space, $\mathcal{A}$ is the action space, $P_s : \mathcal{Z} \times \mathcal{S} \times \mathcal{A} \mapsto \Delta(\mathcal{S})$ is the transition function between states, $P_z : \mathcal{Z} \mapsto \Delta(\mathcal{Z})$ is the transition function between inputs, $R : \mathcal{Z} \times \mathcal{S} \times \mathcal{A} \mapsto [0, R_{\max}]$ is the expected reward function, and $\gamma \in [0, 1)$ is the discount factor (Mao et al., 2018). At each time step $t \in \mathbb{N}_0$ of an IDMDP, the agent observes $S_t$ and $Z_t$, chooses an action $A_t = \pi(S_t, Z_t)$ according to a deterministic policy $\pi : \mathcal{S} \times \mathcal{Z} \to \mathcal{A}$, the next state is drawn as $S_{t+1} \sim P_s(\cdot \mid S_t, A_t, Z_t)$ and the next input comes from $Z_{t+1} \sim P_z(\cdot \mid Z_t)$ which is not impacted by the choice of $A_t$. The agent then receives a reward $R_{t+1}$ using the expected reward function.

**Note 4.1.** *An IDMDP can be rewritten as an MDP with an augmented state space $\bar{\mathcal{S}} = \mathcal{S} \times \mathcal{Z}$ with $P(\bar{S}_{t+1} \mid \bar{S}_t, A_t) = P_z(Z_{t+1} \mid Z_t) P_s(S_{t+1} \mid S_t, A_t)$. Therefore, our decomposition of the commitment loss (Equation 2) and our result on the deliberation cost (Lemma 2.5) still hold with duration-specified options (Definition 2.2).*

As in the classical SMDP, the agent chooses options according to a (deterministic) meta-policy $\mu : \mathcal{S} \times \mathcal{Z} \to \mathcal{O}$ and follow the according intra-option policy until its (possibly random) termination. Each option-selection decision is associated with a deliberation cost given by the

function $C : \mathcal{S} \times \mathcal{Z} \times \mathcal{O} \to [0, C_{\max}]$. Let $T_n$ denote a decision time of the meta-policy. In an IDSMDP, the agent observes the state $S_{T_n}$ and the input $Z_{T_n}$, selects the option $O_n = \mu(S_{T_n}, Z_{T_n})$, incurs a deliberation cost $C(S_{T_n}, Z_{T_n}, O_n)$, and then executes primitive actions according to the intra-option policy $\pi_{O_n} : \mathcal{S} \times \mathcal{Z} \to \mathcal{A}$ until termination of the option.

## 4.2. Approximate Model with Information Drift

We now instantiate an approximate model $\widehat{M}$ of the true underlying IDMDP $M = (\mathcal{S}, \mathcal{Z}, \mathcal{A}, P_s, P_z, R, \gamma)$.

**Definition 4.2** (Frozen-input approximation). Consider the $n$-th option, initiated at time $T_n$ with observed input $z_0 = Z_{T_n}$, and terminating at time $T_{n+1}$ after a (random) duration $K := T_{n+1} - T_n$. During option execution, the primitive state dynamics under $\widehat{M}$ are evaluated using the frozen input $z_0$, i.e.,

$$P_s(\cdot \mid S_t, A_t, z_0) \quad \text{for all } t \in \{T_n, \dots, T_{n+1} - 1\}.$$

At option termination, the agent pays the deliberation cost and observes the true input. The terminal input under $\widehat{M}$ is therefore distributed according to the true $K$–step input transition from $M$:

$$Z_{T_{n+1}} \sim P_z^{(K)}(\cdot \mid z_0),$$

where $P_z^{(k)}(\cdot \mid z)$ denotes the $k$–step transition kernel. Rewards and primitive state dynamics $P_s$ are otherwise identical to those of the true underlying IDMDP $M$.

Note that $\widehat{M}$ is not an IDMDP at the primitive time scale, since the input transition at option boundaries depends on the (random) option duration $K$. It is, however, well-defined through its induced option-level transition kernels. The intuition behind Definition 4.2 is to model scenarios where uncertainty increases with the duration of options, while paying the cost of option termination collapses uncertainty to the true input distribution. The resulting $K$–step input transition is therefore exact, while an accumulating error arises during option execution as a function of input drift. We further assume that both the sensitivity of the policy-induced primitive dynamics to changes in the input and that the rate at which the input can drift over time are bounded.

**Assumption 4.3** (Lipschitz closed-loop dynamics in the input). There exists $L_{\text{dyn}} \geqslant 0$ such that for all states $s \in \mathcal{S}$, inputs $z, z' \in \mathcal{Z}$, and options $o \in \mathcal{O}$,

$$\left\| P_s^{\pi_o}(\cdot \mid s, z) - P_s^{\pi_o}(\cdot \mid s, z') \right\|_1 \leqslant L_{\text{dyn}} \, d_{\mathcal{Z}}(z, z'),$$

where $d_{\mathcal{Z}} : \mathcal{Z} \times \mathcal{Z} \to \mathbb{R}_+$ is a metric on the input space and $P_s^{\pi_o}(\cdot \mid s, z) := P_s(\cdot \mid s, \pi_o(s, z), z)$.

**Assumption 4.4** (Per-step input drift). There exists $\sigma \geqslant 0$ such that the input process satisfies, almost surely,

$$d_{\mathcal{Z}}(Z_{t+1}, Z_t) \leqslant \sigma \quad \text{for all primitive time steps } t.$$

Under the frozen-input approximation model (Definition 4.2), the variance (Equation 2) corresponds to the error due to the model failing to capture the input drift, for which we provide the following upper bound.

**Lemma 4.5** (Drift-Error). *Fix an IDMDP $M = (\mathcal{S}, \mathcal{Z}, \mathcal{A}, P_s, P_z, R, \gamma)$ and an option class $\mathcal{O}_{\mathrm{eval}}$ as in Definition 2.2. Under Assumptions 2.1, 4.3, and 4.4, then the variance (Equation 2) for a frozen-input approximation model $\widehat{M}$ (Definition 4.2) satisfies*

$$\left\| V_{M,\mathcal{O}_{\mathrm{eval}}}^{\mu_{M,\mathcal{O}_{\mathrm{eval}}}^{\star}} - V_{M,\mathcal{O}_{\mathrm{eval}}}^{\mu_{\widehat{M}}^{\star},\mathcal{O}_{\mathrm{eval}}} \right\|_{\infty} \leqslant \frac{2R_{\max}L_{\mathrm{dyn}}\gamma\sigma}{(1-\gamma)^2\beta_{\mathrm{eval}}^2}.$$

The proof relies on perturbation bounds with Assumptions 4.3 and 4.4 to control the divergence between $M$ and $\widehat{M}$ under drifting inputs between options (see Appendix F). When $L_{\mathrm{dyn}}$ and $\sigma$ are large, inputs strongly and rapidly influence state transitions, so long options (low termination probability $\beta_{\mathrm{eval}}$) amplify the variance. Conversely, short options keep the induced policies close, since inputs under $\widehat{M}$ cannot drift too far from those under $M$. This variance reduction comes at an information cost, yielding the following trade-off.

**Theorem 4.6** (Commitment loss from input drift). *Fix an IDMDP $M = (\mathcal{S}, \mathcal{Z}, \mathcal{A}, P_s, P_z, R, \gamma)$ and an option class $\mathcal{O}$ as in Definition 2.2, with $\mathcal{O}_{\mathrm{eval}} \subseteq \mathcal{O}$ induced by the threshold $\beta_{\mathrm{eval}} \in B$. Under Assumptions 2.1, 4.3, and 4.4, then the commitment loss for a frozen-input approximation model $\widehat{M}$ (Definition 4.2) satisfies*

$$\begin{aligned} &\left\| V_{M,\mathcal{O}}^{\mu_{M,\mathcal{O}}^{\star}} - V_{M,\mathcal{O}}^{\mu_{\widehat{M}}^{\star},\mathcal{O}_{\mathrm{eval}}} \right\|_{\infty} \\ &\leqslant \frac{\gamma C_{\max}}{1-\gamma}\left(\beta_{\mathrm{eval}} - \beta_{\min}\right) + \frac{2R_{\max}L_{\mathrm{dyn}}\gamma^2\sigma}{(1-\gamma)^2\beta_{\mathrm{eval}}^2}. \end{aligned}$$

We obtain this result by combining the deliberation-cost bound (Lemma 2.5) with the drift related variance bound (Lemma 4.5) into Equation 2 (see Appendix G). Theorem 4.6 clearly highlights how the cost of information trades off with context-information drift. By solving the bound, we get a recommendation on the optimal option class as a function of the environment's parameters.

**Corollary 4.7** (Bound-optimizing termination rate under input drift). *Under the assumptions of Theorem 4.6, and for fixed $(\gamma, |\mathcal{S}|, |\mathcal{A}|, \delta)$, assume that the minimizer of the bound in Theorem 4.6 lies in the interior $(\beta_{\min}, 1)$. Then*

$$\beta_{\mathrm{eval}}^{\star} = \Theta\left(\left(\frac{R_{\max}L_{\mathrm{dyn}}\sigma}{C_{\max}}\right)^{\frac{1}{3}}\right).$$

*where the hidden constants depend only on $(\gamma, |\mathcal{S}|, |\mathcal{A}|, \delta)$. When the unconstrained minimizer falls outside $[\beta_{\min}, 1]$, the optimal $\beta_{eval}^{\star}$ simply saturates at one of the endpoints $\beta_{\min}$ or $1$.*

The result comes from differentiating the bound with respect to $\beta_{\mathrm{eval}}$ (see Appendix H). The result indicates that when inputs have a high impact over transitions (high $L_{\mathrm{dyn}}$), then the optimal termination rate will become closer to $1$ as the amount of input drift increases (high $\sigma$). Similar to Corollary 3.3, the effect is scaled by the ratio $R_{\max}/C_{\max}$, indicating that information cost influences the optimal termination rate only insofar as it is non-negligible relative to the reward scale of the IDMDP.

# 5. Experiments

We first conduct a simple experiment to support Theorem 3.2 and Corollary 3.3. We then support Theorem 4.6 using experiments in a novel, fast-paced IDMDP environment in which re-planning is costly, but primordial.

## 5.1. Ring IDSMDP

Validating Theorem 3.2 requires an environment for which we can control the shape of $\widehat{M}$. We therefore consider the Ring MDP setting (Jiang et al., 2016). Let $Ring(N, p)$ denote an MDP with $N$ states positioned in the shape of a ring, with every state having two neighboring states. There are two actions, $\mathcal{A} = \{\mathrm{clockwise}, \mathrm{counterclockwise}\}$, which can move the agent to one of the neighbors of the current state. For every pair of non-neighboring states $(s_i, s_j)$ and for each action $a \in \mathcal{A}$, we introduce a transition from $s_i$ to $s_j$ under action $a$ with probability $p$. The remaining transition probabilities over all outgoing edges are then drawn uniformly from $[0, 1]$ and normalized. Similarly, the mean rewards for all state–action pairs are sampled independently from a uniform distribution on $[0, 1]$, which guarantees that $R_{\max} = 1$. We randomly generate 1000 MDPs with $Ring(N = 10, p = 0.125)$ and create approximate models $b$ using maximum likelihood-estimation on $n$ samples of transitions and rewards for each state-action pair.

For each base MDP, we instantiate options using Definition 2.2. We define $B = \{0.02, 0.04, ..., 1\}$ and we use action-repetition options, i.e., $\Pi = \{\pi_a : a \in \mathcal{A}$ with $\pi_a(s) \equiv a$ (Biedenkapp et al., 2021). The deliberation cost $C_{\max}$ is fixed and constant at every option boundary. To control the action space in the resulting SMDP, we consider all values of termination probability threshold $\beta_{\mathrm{eval}} \in B$. For each value of $\beta_{\mathrm{eval}}$, we solve the SMDP using value iteration on the approximate environment $\widehat{M}$ using the corresponding limited option class $\mathcal{O}_{\mathrm{eval}}$ and evaluate the resulting optimal policy on the true $M$. We solve the environment using $\beta_{\min}$ once on the true environment $M$ to get our baseline. We evaluate the impact of option durations on the empirical (normalized) commitment loss:

$$\max_s \left( V_{M,\mathcal{O}}^{\mu_{M,\mathcal{O}}^{\star}}(s) - V_{M,\mathcal{O}}^{\mu_{\widehat{M}}^{\star},\mathcal{O}_{\mathrm{eval}}}(s) \right) / V_{M,\mathcal{O}}^{\mu_{M,\mathcal{O}}^{\star}}(s). \quad (3)$$

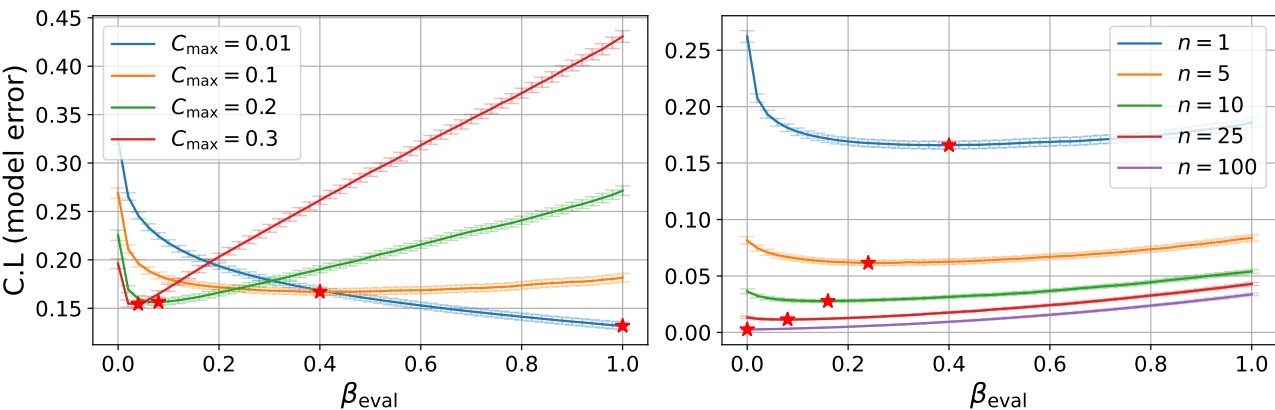

*Figure 1.* Average normalized commitment loss (C.L) (Equation 3) for different deliberation costs (left) and number of samples (right) given $\beta_{\text{eval}}$. The optimal planning horizon is marked by a star.

For each number of samples $n \in \{1, 5, 10, 25, 100\}$, we report the average empirical commitment loss on the 1000 MDP configurations using $C_{\max} = 0.1$ (Figure 1, right). For each delegation cost $C_{\max} \in \{0.01, 0.1, 0.2, 0.3\}$, we report the average empirical commitment loss on the 1000 MDP configurations using $n = 1$ (Figure 1, left).

We can see from Figure 1 that the commitment loss follows the classic U-shape. In the case where the deliberation cost-changes, we see that the higher the cost, the lower the optimal $\beta_{\text{eval}}$ becomes. In the case where the cost is fixed, we see that the lower the amount of samples we have, the higher the optimal termination rate should be (high $\beta_{\text{eval}}$). Both of these results support Corollary 3.3 and are in line with Theorem 3.2.

## 5.2. Cat and Mouse

At a high level, the CatAndMouse($p_{\text{turn}}$) environment instantiates a cat (the learning agent) and a mouse (the target). The goal is for the agent to catch the mouse. More formally, this environment takes place in a $10 \times 10$ grid. The cat observes state $s = (s_{\text{cat}}) \in \mathcal{S}$ and the input $z = (z_{\text{mouse}}, d_{\text{mouse}}) \in \mathcal{Z}$, where $s_{\text{cat}}, z_{\text{mouse}} \in \{0, \ldots, 99\}$ are grid positions and $d_{\text{mouse}} \in \{\uparrow, \rightarrow, \downarrow, \leftarrow\}$ is the mouse's current direction. The action space corresponds to every omnidirectional (8-connected) movement on the grid:

$$\mathcal{A} = \{\uparrow, \downarrow, \leftarrow, \rightarrow, \searcharrow, \nearrow, \swarrow, \searrow\}.$$

We use action-repetition options, i.e., $\Pi = \{\pi_a : a \in \mathcal{A}\}$ with $\pi_a(s) \equiv a$ (Biedenkapp et al., 2021). There are therefore eight intra-option policies. The agent meta-policy chooses from a class of options $\mathcal{O} = \Pi \times B$ where we instantiate $B = \{0.1, 0.2, ..., 1\}$ for a total of $8 \times 10 = 80$ options. Each meta-policy decision incurs a fixed cost $C_{\max}$. There is no cost for decisions made within an option. Once an option is chosen, then environment loops goes like this:

1. The cat executes its action and moves to an adjacent cell (or stays in place if blocked by the boundary) incurring cost of $r_{\text{step}} = -0.2$.

2. If the cat lands on the mouse, the episode terminates with reward $r_{\text{catch}} = 10$.

3. With probability $p_{\text{turn}}$, the mouse selects a new direction from directions that (i) do not step into the cat's trajectory, (ii) move off the cat's trajectory if currently on it, and (iii) increase distance from the cat with priority in that order.

4. The mouse moves one step in its current direction (staying in place if at a boundary).

5. If the cat and mouse occupy the same cell, or swapped positions, the episode terminates with reward $r_{\text{catch}}$.

The loop goes on until the cat chooses an option again. We simulate the conditions of Theorem 4.6 by creating a replay buffer $\widehat{\mathcal{D}} = (s, o, r, c, z_0)$, where $z_0$ is the input seen at the last option boundary (stale information). Buffer $\widehat{\mathcal{D}}$ therefore uses inputs that follow Definition 4.2 of $\widehat{M}$, but gives rewards $r$ that come from $M$ (inputs really do change). By learning an optimal policy in $\widehat{\mathcal{D}}$, we find $\mu_{\widehat{M}, \mathcal{O}}$ that maximizes $V_{M, \mathcal{O}}^{\mu_{\widehat{M}, \mathcal{O}}}$. Our goal is to show that when a learning algorithm has access to all termination probabilities and can validate its performance on the true IDMDP $M$, it will use termination probabilities that matches Corollary 4.7 trading off the cost of information $C_{\max}$, with the drifts of input controlled by $p_{\text{turn}}$. To see if our theory could hold even in modern training regimes, we instantiate this learner as a Deep Q-Network (DQN) (Mnih et al., 2015) to learn the optimal meta-policy.

Using the optimal policy, for each state-input pair in the grid, we look up the optimal termination probability $\beta^\star$, and we

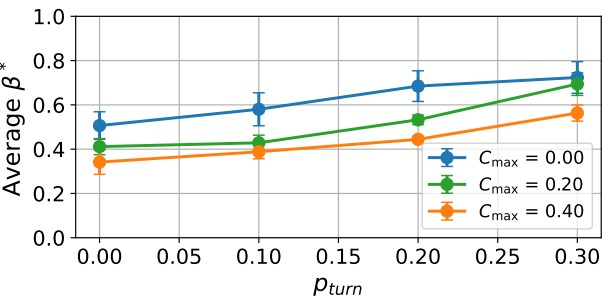

*Figure 2.* Average optimal termination probability across all states and inputs as a function of $p_{\text{turn}}$ for different values of constant deliberation cost $C_{\max}$ in the CatAndMouse environment

take the average of these across all $100 \times 100 \times 4 = 40000$ state-input pairs to get the average optimal termination probability. We run this learner using all $4 \times 3 = 12$ combinations of $p_{\text{turn}} \in \{0, 0.1, 0.2, 0.3\}$ and $C_{\max} \in \{0, 0.2, 0.4\}$ on 5 different seeds[2]. Figure 2 shows the average $\beta^\star$ across 5 seeds with one standard deviation for each configuration.

Figure 2 shows trends consistent with prior work (Harb et al., 2018): as the deliberation cost increases, the DQN selects longer options, resulting in a lower optimal termination probability $\beta^\star$. Beyond this known effect, we observe a systematic interaction with input dynamics. For any fixed deliberation cost, the average optimal $\beta^\star$ increases monotonically with $p_{\text{turn}}$, indicating that faster input drift favors more frequent replanning. Since $p_{\text{turn}}$ serves as a proxy for the input variability parameter $\sigma$ in Corollary 4.7, this behavior is directly predicted by our theoretical analysis.

Together, these results demonstrate that even in more complex, learned control regimes, input drift plays a central role in determining optimal option duration. This empirical study therefore complements Theorem 4.6 and highlights the importance of explicitly modeling input dynamics when reasoning about temporal abstraction.

## 6. Related Work

**Planning horizon in RL**   The dependence on the discount factor in RL is a well studied phenomenon (Jiang et al., 2015; Lefebvre & Durand, 2025; Hu et al., 2022; Jiang et al., 2016). It is well understood that when the credit assigned to rewards far into the future, policy learning on these signal might overfit to what has been seen during training, which brings poor generalization performance (Jiang et al., 2015). In an offline RL scenario, pessimism in the face of uncertainty has been proven to be directly linked with the discount factor (Hu et al., 2022). It is even also well understood how policies generalize better depending on some high level statistics of the environment such as mixing

times or partial observability (Jiang et al., 2016; Lefebvre & Durand, 2025). Despite this huge interest and body of work in the MDP setting, the study of generalization performance in the context of HRL has received almost no attention other than (Harutyunyan et al., 2019). They introduce a new discounting framework by decoupling discount factors for rewards and transitions within options, and derive bias-variance bounds on the resulting estimation error. Our work is complementary: rather than modifying the discounting scheme, we study how option durations interact with model or drift error, and derive bounds on the performance loss due to termination.

**Action Repetition**   A growing body of work studies a restricted class of options corresponding to repeating a single action for a fixed number of time steps (Biedenkapp et al., 2021). Such action repetition has been shown to improve performance and exploration in simple domains (Dabney et al., 2020). However, these methods can fail when the agent enters undesirable states and continues executing a repeated action, leading to large or catastrophic costs. Recent work addresses this issue by explicitly modeling uncertainty (Lee et al., 2024). In particular, Lee et al. (2024) adapt the duration of repeated-action options based on the mean and standard deviation of an ensemble, trading off exploitation and uncertainty via a learned hyperparameter. While effective empirically, this approach lacks theoretical guarantees characterizing when longer or shorter repetitions are preferable. Our results provide grounding on which these algorithms behavior could be explained.

## 7. Conclusion

In this work, we studied how restricting option durations acts as a form of regularization in sequential decision-making problems. We introduced a termination-rate parameter, $\beta_{\text{eval}}$, which monotonically controls the set of admissible option durations, and showed that enforcing shorter options reduces variance arising from both model error (Lemma 3.1) and input drift (Lemma 4.5). However, this regularization effect cannot generally be applied arbitrarily in practice, as interrupting a running option or querying the meta-policy incurs a deliberation cost. This naturally leads to trade-offs between deliberation costs and compounding model or input-induced errors, which we characterize through explicit bounds (Theorems 3.2 and 4.6). Solving these bounds yields principled recommendations for the optimal choice of $\beta_{\text{eval}}$ as a function of the MDP or IDMDP structure (Corollaries 3.3 and 4.7).

**Limitations and future work**   A perceived limitation might be that this work adopts a model-based perspective while most of current hierarchical RL approaches are model-free, as being model-based directly allows us to mathemat-

---

[2]A recap of all hyperparameters can be found in Appendix I.

ically model how errors compound with the commitment length. We must highlight that our insights remain valuable in the model-free case, as the transition model still exists while being implicit (Richens et al., 2025). Another limitation is that our bounds are derived under idealized model-error and input-drift assumptions. Although these abstractions isolate the core trade-offs, extending the analysis to settings with function approximation, correlated data, or learned input models is an important avenue for future work. Finally, our recommendations for $\beta_{\text{eval}}$ give us the optimal threshold for termination probabilities. Although this is very useful to understand the optimal option class to consider in the whole environment, future work could try to provide a theoretically optimal state-dependent probability.

## Acknowledgment

We acknowledge funding from the Canada CIFAR AI Chair program and NSERC Discovery.

## Impact statement

This paper presents work whose goal is to advance the field of Machine Learning. There are many potential societal consequences of our work, none which we feel must be specifically highlighted here.

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

## A. Proof of Equation 2

We want to bound:

$$\left\| V_{M,\mathcal{O}}^{\mu_{M,\mathcal{O}}^\star} - V_{M,\mathcal{O}}^{\mu_{\widehat{M}}^\star,\mathcal{O}_{\text{eval}}} \right\|_\infty$$

To get there, we first need to introduct the following technical result:

**Observation A.1** (Option Evaluation Set Independence). *for any meta policy* $\mu : \mathcal{S} \mapsto \mathcal{O}_{\geqslant \beta_{\text{eval}}}$*, we have:*

$$V_{M,\mathcal{O}}^\mu(s) = V_{M,\mathcal{O}_{\text{eval}}}^\mu(s) \quad \textit{for all } s \in \mathcal{S}.$$

This comes easily from the observation the $\mathcal{O}_{\geqslant \beta_{\text{eval}}} \subseteq \mathcal{O}_{\geqslant \beta_{\min}}$ such that any policy trained using $\mathcal{O}_{\geqslant \beta_{\text{eval}}}$ can be evaluated on $\mathcal{O}_{\geqslant \beta_{\min}}$ with no change in value.

So we have:

$$\begin{aligned}
\left\| V_{M,\mathcal{O}}^{\mu_{M,\mathcal{O}}^\star} - V_{M,\mathcal{O}}^{\mu_{\widehat{M}}^\star,\mathcal{O}_{\text{eval}}} \right\|_\infty &= \left\| V_{M,\mathcal{O}}^{\mu_{M,\mathcal{O}}^\star} - V_{M,\mathcal{O}}^{\mu_{M,\mathcal{O}_{\text{eval}}}^\star} + V_{M,\mathcal{O}}^{\mu_{M,\mathcal{O}_{\text{eval}}}^\star} - V_{M,\mathcal{O}}^{\mu_{\widehat{M}}^\star,\mathcal{O}_{\text{eval}}} \right\|_\infty \\
&\leqslant \left\| V_{M,\mathcal{O}}^{\mu_{M,\mathcal{O}}^\star} - V_{M,\mathcal{O}}^{\mu_{M,\mathcal{O}_{\text{eval}}}^\star} \right\|_\infty + \left\| V_{M,\mathcal{O}_{\text{eval}}}^{\mu_{M,\mathcal{O}_{\text{eval}}}^\star} - V_{M,\mathcal{O}_{\text{eval}}}^{\mu_{\widehat{M}}^\star,\mathcal{O}_{\text{eval}}} \right\|_\infty
\end{aligned}$$

Where we use Observation A.1 and the triangle inequality.

## B. Proof of Lemma 2.5

To upper bound the deliberation cost, we first introduce a shadow meta-policy for the policy acting on $\mathcal{O}_{\text{eval}}$.

**Definition B.1** (Shadow meta-policy). Fix $\beta_{\min}$ and $\beta_{\text{eval}} > \beta_{\min}$, and let $\mu_{M,\mathcal{O}}^\star \colon \mathcal{S} \to \mathcal{O}_{\geqslant \beta_{\min}}$ be an optimal meta-policy for the full option class. For each $s \in \mathcal{S}$, write

$$\mu_{M,\mathcal{O}}^\star(s) = (\pi_s, \beta_s), \quad \text{with } \pi_s \in \Pi, \ \beta_s \in \mathcal{O}.$$

We define the shadow meta-policy $\tilde{\mu}_{M,\mathcal{O}_{\text{eval}}} \colon \mathcal{S} \to \mathcal{O}_{\geqslant \beta_{\text{eval}}}$ by

$$\tilde{\mu}_{M,\mathcal{O}_{\text{eval}}}(s) := \begin{cases} (\pi_s, \beta_s), & \text{if } \beta_s \geqslant \beta_{\text{eval}}, \\ (\pi_s, \beta_{\text{eval}}), & \text{if } \beta_s < \beta_{\text{eval}}. \end{cases}$$

In words, $\tilde{\mu}_{M,\mathcal{O}_{\text{eval}}}$ reuses the same intra-option policy as $\mu_{M,\mathcal{O}}^\star$ at every state, but clamps the termination probability to the admissible range $[\beta_{\text{eval}}, 1]$ by replacing any excessively long option with the longest option that remains admissible under the constraint $\beta \geqslant \beta_{\text{eval}}$.

By optimality of $\mu_{M,\mathcal{O}_{\text{eval}}}^\star$ in $\mathcal{O}_{\geqslant \beta_{\text{eval}}}$, for every $s \in \mathcal{S}$,

$$V_{M,\mathcal{O}_{\text{eval}}}^{\mu_{M,\mathcal{O}_{\text{eval}}}^\star}(s) \geqslant V_{M,\mathcal{O}_{\text{eval}}}^{\tilde{\mu}_{M,\mathcal{O}_{\text{eval}}}}(s),$$

and therefore by using Observation A.1:

$$V_{M,\mathcal{O}}^{\mu_{M,\mathcal{O}_{\text{eval}}}^\star}(s) = V_{M,\mathcal{O}_{\text{eval}}}^{\mu_{M,\mathcal{O}_{\text{eval}}}^\star}(s) \geqslant V_{M,\mathcal{O}_{\text{eval}}}^{\tilde{\mu}_{M,\mathcal{O}_{\text{eval}}}}(s) = V_{M,\mathcal{O}}^{\tilde{\mu}_{M,\mathcal{O}_{\text{eval}}}}(s).$$

Because $\mu_{M,\mathcal{O}}^\star$ is optimal in $\mathcal{O}$, we have

$$V_{M,\mathcal{O}}^{\mu_{M,\mathcal{O}}^\star}(s) \geqslant V_{M,\mathcal{O}}^\mu(s) \quad \text{for every meta-policy } \mu \colon \mathcal{S} \to \mathcal{O}_{\geqslant \beta_{\min}}.$$

In particular, this holds for $\mu = \mu^\star_{M,\mathcal{O}_{\text{eval}}}$ and $\mu = \tilde{\mu}_{M,\mathcal{O}_{\text{eval}}}$. It follows that

$$\left\| V^{\mu^\star_{M,\mathcal{O}}}_{M,\mathcal{O}} - V^{\mu^\star_{M,\mathcal{O}_{\text{eval}}}}_{M,\mathcal{O}} \right\|_\infty \leq \left\| V^{\mu^\star_{M,\mathcal{O}}}_{M,\mathcal{O}} - V^{\tilde{\mu}_{M,\mathcal{O}_{\text{eval}}}}_{M,\mathcal{O}} \right\|_\infty. \tag{4}$$

We now bound the right-hand side. Consider the two SMDP controllers $\mu^\star_{M,\mathcal{O}}$ and $\tilde{\mu}_{M,\mathcal{O}_{\text{eval}}}$ when executing with access to the full option class. We couple their executions on a common probability space as follows:

- they start from the same initial state $S_0 = s$;

- whenever a new option is initiated in state $s$, the two meta-policies select options $(\pi_s, \beta_s)$ and $(\pi_s, \beta'_s)$ respectively, where $\beta'_s := \max\{\beta_s, \beta_{\text{eval}}\}$ as in Definition B.1;

- during execution of an option, both controllers apply the same intra-option policy $\pi_s$ and share the same randomness for environment transitions and rewards; in addition, we generate a common sequence $(U_t)_{t \geq 0}$ of i.i.d. $\text{Unif}[0,1]$ variables and define option termination at time $t$ by

$$D^{\text{base}}_t := \mathbf{1}\{U_t \leq \beta_s\}, \qquad D^{\text{shadow}}_t := \mathbf{1}\{U_t \leq \beta'_s\},$$

where $s$ is the state in which the currently running option was initiated.

Under this coupling, the primitive state–action trajectories $(S_t, A_t)_{t \geq 0}$ and the rewards $(R_{t+1})_{t \geq 0}$ are identical almost surely for both controllers; the only difference lies in the sequence of option initiation times, and hence in the incurred deliberation costs. Moreover, for every $t$ we have $D^{\text{shadow}}_t \geq D^{\text{base}}_t$ almost surely, and $D^{\text{shadow}}_t > D^{\text{base}}_t$ corresponds to an additional early termination of the shadow option, followed by immediate re-initiation of a new option with the same intra-option policy $\pi_s$ in the current state (from Definition B.1).

Let $D^\mu_t := \mathbf{1}\{t = T_n \text{ for some } n\}$ denote the indicator that a new option is initiated at time $t$ under meta-policy $\mu$, and let $C^\mu_t$ be the deliberation cost at time $t$. By boundedness, $0 \leq C^\mu_t \leq C_{\max} D^\mu_t$. Under the coupling, at each time $t$ we have either

$$D^{\text{shadow}}_t = D^{\text{base}}_t \quad \text{and} \quad C^{\text{shadow}}_t = C^{\text{base}}_t,$$

if both options terminated at the same time, or

$$D^{\text{shadow}}_t = 1, \quad D^{\text{base}}_t = 0, \quad \text{and} \quad 0 \leq C^{\text{shadow}}_t - C^{\text{base}}_t \leq C_{\max}.$$

if only the shadow option terminates. Hence, pathwise,

$$0 \leq C^{\text{shadow}}_t - C^{\text{base}}_t \leq C_{\max}\left(D^{\text{shadow}}_t - D^{\text{base}}_t\right).$$

For any initial state $s$,

$$V^{\mu^\star_{M,\mathcal{O}}}_{M,\mathcal{O}}(s) - V^{\tilde{\mu}_{M,\mathcal{O}_{\text{eval}}}}_{M,\mathcal{O}}(s)$$

$$= \mathbb{E}\left[\sum_{t=0}^\infty \gamma^t \left(R_{t+1} - C^{\text{base}}_t\right) \,\Big|\, S_0 = s\right] - \mathbb{E}\left[\sum_{t=0}^\infty \gamma^t \left(R_{t+1} - C^{\text{shadow}}_t\right) \,\Big|\, S_0 = s\right]$$

$$= \mathbb{E}\left[\sum_{t=0}^\infty \gamma^t \left(C^{\text{shadow}}_t - C^{\text{base}}_t\right) \,\Big|\, S_0 = s\right]$$

$$\leq C_{\max} \mathbb{E}\left[\sum_{t=0}^\infty \gamma^t \left(D^{\text{shadow}}_t - D^{\text{base}}_t\right) \,\Big|\, S_0 = s\right]$$

$$= C_{\max} \sum_{t=0}^\infty \gamma^t \left(\mathbb{E}\left[D^{\text{shadow}}_t \,\big|\, S_0 = s\right] - \mathbb{E}\left[D^{\text{base}}_t \,\big|\, S_0 = s\right]\right).$$

At time $t = 0$ both controllers initiate an option, so $D_0^{\text{shadow}} = D_0^{\text{base}} = 1$ and the $t = 0$ term vanishes. Under our assumptions, for $t \geqslant 1$ the initiation indicators are Bernoulli with constant parameters $\beta_{\text{eval}}$ and $\beta_{\text{min}}$ respectively, so

$$\mathbb{E}\big[D_t^{\text{shadow}} \,\big|\, S_0 = s\big] - \mathbb{E}\big[D_t^{\text{base}} \,\big|\, S_0 = s\big] = \beta_{\text{eval}} - \beta_{\text{min}} \quad \text{for all } t \geqslant 1.$$

Thus

$$V_{M,\mathcal{O}}^{\mu_{M,\mathcal{O}}^{\star}}(s) - V_{M,\mathcal{O}}^{\tilde{\mu}_{M,\mathcal{O}_{\text{eval}}}}(s)$$

$$\leqslant C_{\max} \sum_{t=1}^{\infty} \gamma^t \left(\beta_{\text{eval}} - \beta_{\text{min}}\right)$$

$$= C_{\max} \left(\beta_{\text{eval}} - \beta_{\text{min}}\right) \sum_{t=1}^{\infty} \gamma^t$$

$$= C_{\max} \left(\beta_{\text{eval}} - \beta_{\text{min}}\right) \frac{\gamma}{1 - \gamma}$$

$$= \frac{\gamma C_{\max}}{1 - \gamma} \left(\beta_{\text{eval}} - \beta_{\text{min}}\right).$$

Since this bound holds for every $s \in \mathcal{S}$, combining with (4) yields

$$\left\| V_{M,\mathcal{O}}^{\mu_{M,\mathcal{O}}^{\star}} - V_{M,\mathcal{O}}^{\mu_{M,\mathcal{O}_{\text{eval}}}^{\star}} \right\|_{\infty} \leqslant \frac{\gamma C_{\max}}{1 - \gamma} \left(\beta_{\text{eval}} - \beta_{\text{min}}\right). \tag{5}$$

## C. Proof of Lemma 3.1

Recall that the variance term compares the performance of the optimal meta-policy under the true model $M$ and under the approximate model $\widehat{M}$, when both are constrained to the same option class $\mathcal{O}_{\geqslant \beta_{\text{eval}}}$:

$$\left\| V_{M,\mathcal{O}_{\text{eval}}}^{\mu_{M,\mathcal{O}_{\text{eval}}}^{\star}} - V_{M,\mathcal{O}_{\text{eval}}}^{\mu_{\widehat{M},\mathcal{O}_{\text{eval}}}^{\star}} \right\|_{\infty}.$$

For notation convenience, we use $\mu(s,o) = 1_{\mu(s=o)}$. For any meta-policy $\mu : \mathcal{S} \to \mathcal{O}_{\geqslant \beta_{\text{eval}}}$, define

$$\Delta_\mu := \left\| V_{M,\mathcal{O}_{\text{eval}}}^{\mu} - V_{\widehat{M},\mathcal{O}_{\text{eval}}}^{\mu} \right\|_{\infty}.$$

By adding and subtracting $V_{\widehat{M},\mathcal{O}_{\text{eval}}}^{\mu_{M,\mathcal{O}_{\text{eval}}}^{\star}}$ and $V_{M,\mathcal{O}_{\text{eval}}}^{\mu_{\widehat{M},\mathcal{O}_{\text{eval}}}^{\star}}$ and using the triangle inequality, we obtain

$$\left\| V_{M,\mathcal{O}_{\text{eval}}}^{\mu_{M,\mathcal{O}_{\text{eval}}}^{\star}} - V_{M,\mathcal{O}_{\text{eval}}}^{\mu_{\widehat{M},\mathcal{O}_{\text{eval}}}^{\star}} \right\|_{\infty} \leqslant \left\| V_{M,\mathcal{O}_{\text{eval}}}^{\mu_{M,\mathcal{O}_{\text{eval}}}^{\star}} - V_{\widehat{M},\mathcal{O}_{\text{eval}}}^{\mu_{M,\mathcal{O}_{\text{eval}}}^{\star}} \right\|_{\infty} + \left\| V_{M,\mathcal{O}_{\text{eval}}}^{\mu_{\widehat{M},\mathcal{O}_{\text{eval}}}^{\star}} - V_{\widehat{M},\mathcal{O}_{\text{eval}}}^{\mu_{\widehat{M},\mathcal{O}_{\text{eval}}}^{\star}} \right\|_{\infty}$$

$$\leqslant 2 \max_\mu \Delta_\mu.$$

Thus it is enough to bound $\max_\mu \Delta_\mu$.

Fix an arbitrary meta-policy $\mu$ and let

$$V := V_{M,\mathcal{O}_{\text{eval}}}^{\mu}, \qquad \widehat{V} := V_{\widehat{M},\mathcal{O}_{\text{eval}}}^{\mu}.$$

For each option $o$ and state $s$, let $K \geqslant 1$ denote the (random) primitive duration of the option when initiated at $s$, and let $S_{t+K}$ be its termination state (similar to (Sutton et al., 1999)). Define the option-level reward and discounted transition under $M$ as

$$r_M^o(s) := \mathbb{E}\left[ \sum_{i=0}^{K-1} \gamma^i \left( R_{t+1+i} - C_{t+i} \right) \,\middle|\, S_t = s,\ O_t = o \right],$$

$$p_M^o(s'|s) := \sum_{k=1}^{\infty} \gamma^k \,\mathbb{P}\big( S_{t+k} = s',\ K = k \,\big|\, S_t = s,\ O_t = o \big), \qquad s' \in \mathcal{S}.$$

Where we abuse the notation and remove the $K$ from the joint contribution for space. $P\big(\bar{S}_{t+k} = \bar{s}' \,\big|\, \bar{S}_t = \bar{s},\, O_t = o\big)$ is the $k$-step kernel from executing the intra option policy $\pi_o$ for $k$ steps. $A_{ab}$ is the entry in row $a$, column $b$ of matrix $A$. Given a deterministic meta-policy $\mu$, aggregate these into

$$R_M^\mu(s) := \sum_{o \in \mathcal{O}_{\geqslant \beta_{\text{eval}}}} \mu(s, o)\, r_M^o(s),$$

$$P_M^\mu(s, s') := \sum_{o \in \mathcal{O}_{\geqslant \beta_{\text{eval}}}} \mu(s, o)\, p_M^o(s'|s), \qquad s, s' \in \mathcal{S}.$$

We view $R_M^\mu$ as a vector in $\mathbb{R}^{|\mathcal{S}|}$ and $P_M^\mu$ as a matrix in $\mathbb{R}^{|\mathcal{S}| \times |\mathcal{S}|}$. By construction,

$$\sum_{s'} P_M^\mu(s, s') = \sum_o \mu(s, o) \sum_{s'} p_M^o(s'|s) = \sum_o \mu(s, o)\, \mathbb{E}[\gamma^K \mid S_t = s, O_t = o] \leqslant \gamma,$$

since $K \geqslant 1$ almost surely and $\gamma \in [0, 1)$.

Define the SMDP Bellman operator induced by $\mu$ under $M$ as

$$(T^\mu V)(s) := R_M^\mu(s) + \sum_{s'} P_M^\mu(s, s') V(s'),$$

and analogously under $\widehat{M}$ with $R_{\widehat{M}}^\mu$ and $P_{\widehat{M}}^\mu$, yielding $\widehat{T}^\mu$. By the bound on the row sums of $P_M^\mu$, $T^\mu$ is a $\gamma$-contraction in the sup-norm:

$$\|T^\mu V - T^\mu W\|_\infty \leqslant \gamma \|V - W\|_\infty \quad \text{for all } V, W.$$

By definition of these operators, the value functions satisfy

$$V = T^\mu V, \qquad \widehat{V} = \widehat{T}^\mu \widehat{V}.$$

We now bound $\Delta_\mu$ via a standard perturbation argument:

$$\begin{aligned}
\Delta_\mu = \|V - \widehat{V}\|_\infty &= \|T^\mu V - \widehat{T}^\mu \widehat{V}\|_\infty \\
&= \|T^\mu V - T^\mu \widehat{V} + T^\mu \widehat{V} - \widehat{T}^\mu \widehat{V}\|_\infty \\
&\leqslant \|T^\mu V - T^\mu \widehat{V}\|_\infty + \|(T^\mu - \widehat{T}^\mu)\widehat{V}\|_\infty \\
&\leqslant \gamma \|V - \widehat{V}\|_\infty + \|T^\mu - \widehat{T}^\mu\|_\infty \, \|\widehat{V}\|_\infty \\
&= \gamma \Delta_\mu + \|T^\mu - \widehat{T}^\mu\|_\infty \, \|\widehat{V}\|_\infty.
\end{aligned}$$

Where $\|T^\mu - \widehat{T}^\mu\|_\infty$ is a matrix because rewards are identical under $M$ and $\widehat{M}$ (see section 1) and the matrix sup-norm, $\|A\|_\infty := \max_s \sum_{s'} |A(s, s')|$ which is the highest row sum. Rearranging gives

$$(1 - \gamma)\Delta_\mu \leqslant \|T^\mu - \widehat{T}^\mu\|_\infty \, \|\widehat{V}\|_\infty.$$

Using the bounded reward assumption, we have

$$\|\widehat{V}\|_\infty \leqslant \frac{R_{\max}}{1 - \gamma},$$

so

$$\Delta_\mu \leqslant \frac{R_{\max}}{(1 - \gamma)^2} \|T^\mu - \widehat{T}^\mu\|_\infty.$$

Taking the maximum over $\mu$ yields the intermediate bound

$$\max_\mu \big\| V_{M, \mathcal{O}_{\text{eval}}}^\mu - V_{\widehat{M}, \mathcal{O}_{\text{eval}}}^\mu \big\|_\infty \leqslant \frac{R_{\max}}{(1 - \gamma)^2} \max_\mu \big\| T^\mu - \widehat{T}^\mu \big\|_\infty.$$

So now we can focus on the difference between the SMDP operators. Since rewards are identical in $M$ and $\widehat{M}$, the affine parts cancel and

$$\|T^\mu - \widehat{T}^\mu\|_\infty = \|P_M^\mu - P_{\widehat{M}}^\mu\|_\infty.$$

Introduce $e_s^\top \in \mathbb{R}^{|\mathcal{S}|}$ as the row vector which is 1 in coordinate $s$ and 0 elsewhere. By definition of the matrix $\infty$–norm,

$$\|P_M^\mu - P_{\widehat{M}}^\mu\|_\infty = \max_s \|e_s^\top (P_M^\mu - P_{\widehat{M}}^\mu)\|_1.$$

Fix $s \in \mathcal{S}$ and write $\mu(s) = o_s = (\pi_s, \beta_s)$ with $\beta_s \in \mathcal{O}_{\text{eval}}$. Under Assumption 2.1, the multi-time option kernel at $s$ can be written as

$$e_s^\top P_M^\mu = \sum_{n=1}^\infty \gamma^n \beta_s (1 - \beta_s)^{n-1} e_s^\top (P_M^{\pi_s})^n, \qquad e_s^\top P_{\widehat{M}}^\mu = \sum_{n=1}^\infty \gamma^n \beta_s (1 - \beta_s)^{n-1} e_s^\top (P_{\widehat{M}}^{\pi_s})^n,$$

where $P_M^{\pi_s}$ and $P_{\widehat{M}}^{\pi_s}$ are the primitive transition matrices under $\pi_s$ in $M$ and $\widehat{M}$, respectively. For brevity, write $A := P_M^{\pi_s}$ and $B := P_{\widehat{M}}^{\pi_s}$. Then

$$e_s^\top (P_M^\mu - P_{\widehat{M}}^\mu) = \sum_{n=1}^\infty \gamma^n \beta_s (1 - \beta_s)^{n-1} e_s^\top (A^n - B^n).$$

Using the identity $A^n - B^n = \sum_{k=0}^{n-1} A^k (A - B) B^{n-1-k}$, we obtain

$$e_s^\top (P_M^\mu - P_{\widehat{M}}^\mu) = \sum_{n=1}^\infty \gamma^n \beta_s (1 - \beta_s)^{n-1} \sum_{k=0}^{n-1} e_s^\top A^k (A - B) B^{n-1-k}.$$

Taking the $\ell_1$–norm and applying the triangle inequality yields

$$\|e_s^\top (P_M^\mu - P_{\widehat{M}}^\mu)\|_1 \leqslant \sum_{n=1}^\infty \gamma^n \beta_s (1 - \beta_s)^{n-1} \sum_{k=0}^{n-1} \|e_s^\top A^k (A - B) B^{n-1-k}\|_1.$$

Since $A$ and $B$ are row-stochastic, $A^k$ and $B^{n-1-k}$ are also row-stochastic, and hence

$$\|A^k\|_\infty = \|B^{n-1-k}\|_\infty = 1.$$

Moreover, for any probability row vector $q$ and any matrix $M$,

$$\|qM\|_1 \leqslant \max_i \|e_i^\top M\|_1 = \|M\|_\infty.$$

Applying this with $q = e_s^\top A^k$ and $M = (A - B) B^{n-1-k}$ gives

$$\|e_s^\top A^k (A - B) B^{n-1-k}\|_1 \leqslant \|(A - B) B^{n-1-k}\|_\infty.$$

By submultiplicativity of the $\infty$–norm,

$$\|(A - B) B^{n-1-k}\|_\infty \leqslant \|A - B\|_\infty \|B^{n-1-k}\|_\infty = \|A - B\|_\infty,$$

so

$$\|e_s^\top A^k (A - B) B^{n-1-k}\|_1 \leqslant \|A - B\|_\infty = \|P_M^{\pi_s} - P_{\widehat{M}}^{\pi_s}\|_\infty.$$

Therefore,

$$\|e_s^\top (P_M^\mu - P_{\widehat{M}}^\mu)\|_1 \leqslant \sum_{n=1}^\infty \gamma^n \beta_s (1 - \beta_s)^{n-1} n \|P_M^{\pi_s} - P_{\widehat{M}}^{\pi_s}\|_\infty.$$

Evaluating the geometric series with $r := \gamma(1 - \beta_s)$ gives

$$\sum_{n=1}^\infty \gamma^n \beta_s (1 - \beta_s)^{n-1} n = \beta_s \gamma \sum_{n=1}^\infty n r^{n-1} = \frac{\beta_s \gamma}{\left(1 - \gamma(1 - \beta_s)\right)^2},$$

so

$$\left\| e_s^\top (P_M^\mu - P_{\widehat{M}}^\mu) \right\|_1 \leqslant \frac{\beta_s \gamma}{\left(1 - \gamma(1 - \beta_s)\right)^2} \|P_M^{\pi_s} - P_{\widehat{M}}^{\pi_s}\|_\infty.$$

Because $\beta_s \in \mathcal{O}_{\mathrm{eval}}$, we have $\beta_s \geqslant \beta_{\mathrm{eval}}$. Moreover

$$1 - \gamma(1 - \beta_s) = (1 - \gamma) + \gamma\beta_s \geqslant \beta_s,$$

since $(1 - \gamma)(1 - \beta_s) \geqslant 0$. Therefore

$$\left(1 - \gamma(1 - \beta_s)\right)^2 \geqslant \beta_s^2, \qquad \frac{\beta_s}{\left(1 - \gamma(1 - \beta_s)\right)^2} \leqslant \frac{1}{\beta_s} \leqslant \frac{1}{\beta_{\mathrm{eval}}}.$$

Multiplying by $\gamma$ and using $\beta_{\mathrm{eval}} \leqslant 1$ yields

$$\frac{\beta_s \gamma}{\left(1 - \gamma(1 - \beta_s)\right)^2} \leqslant \frac{\gamma}{\beta_{\mathrm{eval}}}$$

Therefore,

$$\left\| e_s^\top (P_M^\mu - P_{\widehat{M}}^\mu) \right\|_1 \leqslant \frac{\gamma}{\beta_{\mathrm{eval}}} \|P_M^{\pi_s} - P_{\widehat{M}}^{\pi_s}\|_\infty.$$

Taking the maximum over $s$ and over all possible intra-option policies $\pi_s \in \Pi$ yields

$$\|P_M^\mu - P_{\widehat{M}}^\mu\|_\infty = \max_s \left\| e_s^\top (P_M^\mu - P_{\widehat{M}}^\mu) \right\|_1 \leqslant \frac{\gamma}{\beta_{\mathrm{eval}}} \max_{\pi \in \Pi} \|P_M^\pi - P_{\widehat{M}}^\pi\|_\infty.$$

Combining with the intermediate bound

$$\max_\mu \|V_{M,\mathcal{O}_{\mathrm{eval}}}^\mu - V_{\widehat{M},\mathcal{O}_{\mathrm{eval}}}^\mu\|_\infty \leqslant \frac{R_{\max}}{(1 - \gamma)^2} \max_\mu \|T^\mu - \widehat{T}^\mu\|_\infty = \frac{R_{\max}}{(1 - \gamma)^2} \max_\mu \|P_M^\mu - P_{\widehat{M}}^\mu\|_\infty$$

and the factor 2 from the reduction to fixed meta-policies gives

$$\left\| V_{M,\mathcal{O}_{\mathrm{eval}}}^{\mu_{M,\mathcal{O}_{\mathrm{eval}}}^\star} - V_{M,\mathcal{O}_{\mathrm{eval}}}^{\mu_{\widehat{M},\mathcal{O}_{\mathrm{eval}}}^\star} \right\|_\infty \leqslant \frac{2\gamma R_{\max}}{(1 - \gamma)^2 \beta_{\mathrm{eval}}} \max_{\pi \in \Pi} \|P_M^\pi - P_{\widehat{M}}^\pi\|_\infty.$$

It remains to control the one-step model error $\max_{\pi \in \Pi} \|P_M^\pi - P_{\widehat{M}}^\pi\|_\infty$. Recall that $\widehat{M}$ is obtained by maximum-likelihood estimation of each primitive kernel $P(\cdot \mid s, a)$ from $n$ i.i.d. samples per state–action pair $(s, a)$. Let $\hat{P}(\cdot \mid s, a)$ denote the empirical distribution. For any fixed $(s, a)$, (Weissman et al., 2003) showed that for all $\varepsilon > 0$,

$$\mathbb{P}\big(\|\hat{P}(\cdot \mid s, a) - P(\cdot \mid s, a)\|_1 \geqslant \varepsilon\big) \leqslant (2^{|\mathcal{S}|} - 2) \exp\left(-\frac{n\varepsilon^2}{2}\right).$$

Applying a union bound over all $(s, a) \in \mathcal{S} \times \mathcal{A}$ and solving for $\varepsilon$ yields that with probability at least $1 - \delta$,

$$\max_{s,a} \|\hat{P}(\cdot \mid s, a) - P(\cdot \mid s, a)\|_1 \leqslant \sqrt{\frac{2}{n}\left(\log(2^{|\mathcal{S}|} - 2) + \log\frac{|\mathcal{S}||\mathcal{A}|}{\delta}\right)}.$$

For any deterministic policy $\pi$, we have

$$\|P_M^\pi - P_{\widehat{M}}^\pi\|_\infty = \max_s \|P(\cdot \mid s, \pi(s)) - \hat{P}(\cdot \mid s, \pi(s))\|_1 \leqslant \max_{s,a} \|\hat{P}(\cdot \mid s, a) - P(\cdot \mid s, a)\|_1,$$

so the same bound holds for $\max_{\pi \in \Pi} \|P_M^\pi - P_{\widehat{M}}^\pi\|_\infty$. Plugging this into the previous inequality, we conclude that with probability at least $1 - \delta$,

$$\left\| V_{M,\mathcal{O}_{\mathrm{eval}}}^{\mu_{M,\mathcal{O}_{\mathrm{eval}}}^\star} - V_{M,\mathcal{O}_{\mathrm{eval}}}^{\mu_{\widehat{M},\mathcal{O}_{\mathrm{eval}}}^\star} \right\|_\infty \leqslant \frac{2\gamma R_{\max}}{(1 - \gamma)^2 \beta_{\mathrm{eval}}} \sqrt{\frac{2}{n}\left(\log(2^{|\mathcal{S}|} - 2) + \log\frac{|\mathcal{S}||\mathcal{A}|}{\delta}\right)}.$$

This establishes the claimed variance bound.

# D. Proof of Theorem 3.2

Recall Equation 2:

$$\underbrace{\left\|V_{M,\mathcal{O}}^{\mu_{M,\mathcal{O}}^{\star}} - V_{M,\mathcal{O}}^{\mu_{\widehat{M}}^{\star},\mathcal{O}_{\text{eval}}}\right\|_{\infty}}_{\text{commitment loss}} \leqslant \underbrace{\left\|V_{M,\mathcal{O}}^{\mu_{M,\mathcal{O}}^{\star}} - V_{M,\mathcal{O}}^{\mu_{M}^{\star},\mathcal{O}_{\text{eval}}}\right\|_{\infty}}_{\text{deliberation cost}} + \underbrace{\left\|V_{M,\mathcal{O}_{\text{eval}}}^{\mu_{M}^{\star},\mathcal{O}_{\text{eval}}} - V_{M,\mathcal{O}_{\text{eval}}}^{\mu_{\widehat{M}}^{\star},\mathcal{O}_{\text{eval}}}\right\|_{\infty}}_{\text{variance}} \tag{6}$$

$$\tag{7}$$

We use Lemma 2.5 and Lemma 3.1 to get:

$$\underbrace{\left\|V_{M,\mathcal{O}}^{\mu_{M,\mathcal{O}}^{\star}} - V_{M,\mathcal{O}}^{\mu_{\widehat{M}}^{\star},\mathcal{O}_{\text{eval}}}\right\|_{\infty}}_{\text{commitment loss}} \leqslant \underbrace{\left\|V_{M,\mathcal{O}}^{\mu_{M,\mathcal{O}}^{\star}} - V_{M,\mathcal{O}}^{\mu_{M}^{\star},\mathcal{O}_{\text{eval}}}\right\|_{\infty}}_{\text{deliberation cost}} + \underbrace{\left\|V_{M,\mathcal{O}_{\text{eval}}}^{\mu_{M}^{\star},\mathcal{O}_{\text{eval}}} - V_{M,\mathcal{O}_{\text{eval}}}^{\mu_{\widehat{M}}^{\star},\mathcal{O}_{\text{eval}}}\right\|_{\infty}}_{\text{variance}} \tag{8}$$

$$\leqslant \frac{\gamma C_{\max}}{1-\gamma}\left(\beta_{\text{eval}} - \beta_{\min}\right) + \frac{2\gamma R_{\max}}{(1-\gamma)^2 \beta_{\text{eval}}}\sqrt{\frac{2}{n}\left(\log(2^{|\mathcal{S}|}-2) + \log\frac{|\mathcal{S}||\mathcal{A}|}{\delta}\right)}. \tag{9}$$

# E. Proof of Corollary 4.7

By Theorem 3.2, for any $\beta_{\text{eval}} \in [\beta_{\min}, 1]$ we have

$$\left\|V_{M,\mathcal{O}}^{\mu_{M,\mathcal{O}}^{\star}} - V_{M,\mathcal{O}}^{\mu_{\widehat{M}}^{\star},\mathcal{O}_{\text{eval}}}\right\|_{\infty} \leqslant \frac{\gamma C_{\max}}{1-\gamma}\left(\beta_{\text{eval}} - \beta_{\min}\right) + \frac{2\gamma R_{\max}}{(1-\gamma)^2 \beta_{\text{eval}}}\sqrt{\frac{2}{n}\left(\log(2^{|\mathcal{S}|}-2) + \log\frac{|\mathcal{S}||\mathcal{A}|}{\delta}\right)}$$

$$= -A\beta_{\min} + A\beta_{\text{eval}} + \frac{B}{\beta_{\text{eval}}},$$

where

$$A := \frac{\gamma C_{\max}}{1-\gamma}, \qquad B := \frac{2\gamma R_{\max}}{(1-\gamma)^2}\sqrt{\frac{2}{n}\left(\log(2^{|\mathcal{S}|}-2) + \log\frac{|\mathcal{S}||\mathcal{A}|}{\delta}\right)}.$$

The term $-A\beta_{\min}$ does not depend on $\beta_{\text{eval}}$, so minimizing the bound over $\beta_{\text{eval}} \in [\beta_{\min}, 1]$ is equivalent to minimizing

$$g(\beta) := A\beta + \frac{B}{\beta}, \qquad \beta \in [\beta_{\min}, 1].$$

Since $A > 0$ and $B > 0$, $g$ is differentiable on $(0, \infty)$ with

$$g'(\beta) = A - \frac{B}{\beta^2}, \qquad g''(\beta) = \frac{2B}{\beta^3} > 0 \quad \text{for all } \beta > 0.$$

Thus $g$ is strictly convex on $(0, \infty)$ and has at most one critical point. Solving $g'(\beta) = 0$ yields

$$A - \frac{B}{\beta^2} = 0 \quad \Longrightarrow \quad \beta^2 = \frac{B}{A} \quad \Longrightarrow \quad \beta_0 := \sqrt{\frac{B}{A}}.$$

Because $g''(\beta_0) > 0$, this critical point is the unique global minimizer of $g$ on $(0, \infty)$.

We now restrict to the feasible interval $[\beta_{\min}, 1]$. There are three cases:

1. If $\beta_0 \in [\beta_{\min}, 1]$, then by convexity the minimum of $g$ on $[\beta_{\min}, 1]$ is attained at $\beta_0$.

2. If $\beta_0 < \beta_{\min}$, then $g'(\beta) > 0$ for all $\beta \geqslant \beta_{\min}$, so $g$ is strictly increasing on $[\beta_{\min}, 1]$ and the minimum is attained at $\beta_{\min}$.

3. If $\beta_0 > 1$, then $g'(\beta) < 0$ for all $\beta \leqslant 1$, so $g$ is strictly decreasing on $[\beta_{\min}, 1]$ and the minimum is attained at 1.

In all cases, the minimizer over $[\beta_{\min}, 1]$ is given by

$$\beta_{\text{eval}}^{\star} = \text{clip}_{[\beta_{\min}, 1]}(\beta_0) = \text{clip}_{[\beta_{\min}, 1]}\left(\sqrt{\frac{B}{A}}\right),$$

where we define the clipping operator

$$\text{clip}_{[\beta_{\min}, 1]}(x) := \min\{1, \max\{\beta_{\min}, x\}\}.$$

Finally, observing that $A \propto C_{\max}$ and $B \propto R_{\max}\sqrt{\Lambda/n}$ with

$$\Lambda := \log(2^{|\mathcal{S}|} - 2) + \log\frac{|\mathcal{S}||\mathcal{A}|}{\delta},$$

we obtain, for fixed $(\gamma, |\mathcal{S}|, |\mathcal{A}|, \delta)$,

$$\beta_{\text{eval}}^{\star} = \Theta\left(\sqrt{\frac{R_{\max}}{C_{\max}}}\, n^{-1/4}\right).$$

## F. Proof of Lemma 4.5

Recall that the variance term compares the performance of the optimal meta-policy under the true model $M$ and under the approximate model $\widehat{M}$, when both are constrained to the same option class $\mathcal{O}_{\geqslant \beta_{\text{eval}}}$ on the augmented state space $\bar{\mathcal{S}} := \mathcal{S} \times \mathcal{Z}$:

$$\left\| V_{M, \mathcal{O}_{\text{eval}}}^{\mu_{M, \mathcal{O}_{\text{eval}}}^{\star}} - V_{M, \mathcal{O}_{\text{eval}}}^{\mu_{\widehat{M}, \mathcal{O}_{\text{eval}}}^{\star}} \right\|_{\infty}.$$

For any meta-policy $\mu : \bar{\mathcal{S}} \to \mathcal{O}_{\geqslant \beta_{\text{eval}}}$. For notation convenience, we use $\mu(o|\bar{s}) = 1_{\mu(\bar{s}=o)}$ define

$$\Delta_{\mu} := \left\| V_{M, \mathcal{O}_{\text{eval}}}^{\mu} - V_{\widehat{M}, \mathcal{O}_{\text{eval}}}^{\mu} \right\|_{\infty}.$$

By adding and subtracting $V_{\widehat{M}, \mathcal{O}_{\text{eval}}}^{\mu_{M, \mathcal{O}_{\text{eval}}}^{\star}}$ and $V_{M, \mathcal{O}_{\text{eval}}}^{\mu_{\widehat{M}, \mathcal{O}_{\text{eval}}}^{\star}}$ and using the triangle inequality, we obtain

$$\left\| V_{M, \mathcal{O}_{\text{eval}}}^{\mu_{M, \mathcal{O}_{\text{eval}}}^{\star}} - V_{M, \mathcal{O}_{\text{eval}}}^{\mu_{\widehat{M}, \mathcal{O}_{\text{eval}}}^{\star}} \right\|_{\infty} \leqslant \left\| V_{M, \mathcal{O}_{\text{eval}}}^{\mu_{M, \mathcal{O}_{\text{eval}}}^{\star}} - V_{\widehat{M}, \mathcal{O}_{\text{eval}}}^{\mu_{M, \mathcal{O}_{\text{eval}}}^{\star}} \right\|_{\infty} + \left\| V_{M, \mathcal{O}_{\text{eval}}}^{\mu_{\widehat{M}, \mathcal{O}_{\text{eval}}}^{\star}} - V_{\widehat{M}, \mathcal{O}_{\text{eval}}}^{\mu_{\widehat{M}, \mathcal{O}_{\text{eval}}}^{\star}} \right\|_{\infty}$$

$$\leqslant 2 \max_{\mu} \Delta_{\mu}.$$

Thus it is enough to bound $\max_{\mu} \Delta_{\mu}$.

Fix an arbitrary meta-policy $\mu$ and let

$$V := V_{M, \mathcal{O}_{\text{eval}}}^{\mu}, \qquad \widehat{V} := V_{\widehat{M}, \mathcal{O}_{\text{eval}}}^{\mu},$$

where $V$ and $\widehat{V}$ are defined on $\bar{\mathcal{S}} = \mathcal{S} \times \mathcal{Z}$.

For each option $o$ and augmented state $\bar{s} = (s, z)$, let $K \geqslant 1$ denote the (random) primitive duration of the option when initiated at $\bar{s}$, and let $\bar{S}_{t+K}$ be its termination augmented state. As in the standard options framework (Sutton et al., 1999), we define the option-level (discounted) reward and transition under $M$ as

$$r_M^o(\bar{s}) := \mathbb{E}\left[ \sum_{i=0}^{K-1} \gamma^i \left( R_{t+1+i} - C_{t+i} \right) \,\middle|\, \bar{S}_t = \bar{s},\, O_t = o \right],$$

$$p_M^o(\bar{s}'|\bar{s}) := \sum_{k=1}^{\infty} \gamma^k \mathbb{P}\left( \bar{S}_{t+k} = \bar{s}',\, K = k \,\middle|\, \bar{S}_t = \bar{s},\, O_t = o \right), \qquad \bar{s}' \in \bar{\mathcal{S}}.$$

$$= \sum_{k=1}^{\infty} \gamma^k \mathbb{P}\left( \bar{S}_{t+k} = \bar{s}' \,\middle|\, \bar{S}_t = \bar{s},\, O_t = o \right), \qquad \bar{s}' \in \bar{\mathcal{S}}.$$

Where we abuse the notation and remove the $K$ from the joint contribution for space. $P\big(\bar{S}_{t+k} = \bar{s}' \,|\, \bar{S}_t = \bar{s},\ O_t = o\big)$ is the $k$-step kernel from executing the intra option policy $\pi_o$ for $k$ steps. Given a deterministic meta-policy $\mu$ on $\bar{\mathcal{S}}$, aggregate these into

$$R_M^\mu(\bar{s}) := \sum_{o \in \mathcal{O}_{\geqslant \beta_{\mathrm{eval}}}} \mu(o|\bar{s})\, r_M^o(\bar{s}),$$

$$P_M^\mu(\bar{s}, \bar{s}') := \sum_{o \in \mathcal{O}_{\geqslant \beta_{\mathrm{eval}}}} \mu(o|\bar{s})\, p_M^o(\bar{s}'|\bar{s}), \qquad \bar{s}, \bar{s}' \in \bar{\mathcal{S}}.$$

We view $R_M^\mu$ as a vector in $\mathbb{R}^{|\bar{\mathcal{S}}|}$ and $P_M^\mu$ as a matrix in $\mathbb{R}^{|\bar{\mathcal{S}}| \times |\bar{\mathcal{S}}|}$. By construction, and using state-independent termination (Assumption 2.1),

$$\sum_{\bar{s}'} P_M^\mu(\bar{s}, \bar{s}') = \sum_o \mu(o|\bar{s}) \sum_{\bar{s}'} p_M^o(\bar{s}'|\bar{s}) = \sum_o \mu(o|\bar{s})\, \mathbb{E}[\gamma^K \mid \bar{S}_t = \bar{s}, O_t = o] \leqslant \gamma,$$

since $K \geqslant 1$ almost surely and $\gamma \in [0, 1)$.

Define the SMDP Bellman operator induced by $\mu$ under $M$ as

$$(T^\mu V)(\bar{s}) := R_M^\mu(\bar{s}) + \sum_{\bar{s}'} P_M^\mu(\bar{s}, \bar{s}') V(\bar{s}'),$$

and analogously under $\widehat{M}$ with $R_{\widehat{M}}^\mu$ and $P_{\widehat{M}}^\mu$, yielding $\widehat{T}^\mu$. By the bound on the row sums of $P_M^\mu$, $T^\mu$ is a $\gamma$-contraction in the sup-norm:

$$\|T^\mu V - T^\mu W\|_\infty \leqslant \gamma \|V - W\|_\infty \quad \text{for all } V, W.$$

By definition of these operators, the value functions satisfy

$$V = T^\mu V, \qquad \widehat{V} = \widehat{T}^\mu \widehat{V}.$$

We now bound $\Delta_\mu$ via a standard perturbation argument:

$$\begin{aligned}
\Delta_\mu = \|V - \widehat{V}\|_\infty &= \|T^\mu V - \widehat{T}^\mu \widehat{V}\|_\infty \\
&= \|T^\mu V - T^\mu \widehat{V} + T^\mu \widehat{V} - \widehat{T}^\mu \widehat{V}\|_\infty \\
&\leqslant \|T^\mu V - T^\mu \widehat{V}\|_\infty + \|(T^\mu - \widehat{T}^\mu)\widehat{V}\|_\infty \\
&\leqslant \gamma \|V - \widehat{V}\|_\infty + \|T^\mu - \widehat{T}^\mu\|_\infty \|\widehat{V}\|_\infty \\
&= \gamma \Delta_\mu + \|T^\mu - \widehat{T}^\mu\|_\infty \|\widehat{V}\|_\infty.
\end{aligned}$$

Here $\|T^\mu - \widehat{T}^\mu\|_\infty$ is the induced matrix norm (since rewards are identical under $M$ and $\widehat{M}$; see Section 1) and the matrix sup-norm is defined by $\|A\|_\infty := \max_i \sum_j |A_{ij}|$. Rearranging gives

$$(1 - \gamma)\Delta_\mu \leqslant \|T^\mu - \widehat{T}^\mu\|_\infty \|\widehat{V}\|_\infty.$$

Using the bounded reward assumption, we have

$$\|\widehat{V}\|_\infty \leqslant \frac{R_{\max}}{1 - \gamma},$$

so

$$\Delta_\mu \leqslant \frac{R_{\max}}{(1 - \gamma)^2} \|T^\mu - \widehat{T}^\mu\|_\infty.$$

Taking the maximum over $\mu$ yields the intermediate bound

$$\max_\mu \big\| V_{M, \mathcal{O}_{\mathrm{eval}}}^\mu - V_{\widehat{M}, \mathcal{O}_{\mathrm{eval}}}^\mu \big\|_\infty \leqslant \frac{R_{\max}}{(1 - \gamma)^2} \max_\mu \|T^\mu - \widehat{T}^\mu\|_\infty. \tag{10}$$

Since rewards are identical in $M$ and $\widehat{M}$, the reward parts cancel and

$$\|T^\mu - \widehat{T}^\mu\|_\infty = \|P_M^\mu - P_{\widehat{M}}^\mu\|_\infty.$$

Introduce $e_{\bar{s}}^\top \in \mathbb{R}^{|\bar{\mathcal{S}}|}$ as the row vector which is 1 in coordinate $\bar{s}$ and 0 elsewhere. By definition of the matrix $\infty$–norm,

$$\|P_M^\mu - P_{\widehat{M}}^\mu\|_\infty = \max_{\bar{s}}\|e_{\bar{s}}^\top (P_M^\mu - P_{\widehat{M}}^\mu)\|_1.$$

Fix $\bar{s}_0 = (s_0, z_0), \in \bar{\mathcal{S}}$, from now on, we refer to the chosen option as $o = \mu(\bar{s}_0)$ with $(\pi_o, \beta_o)$ the corresponding intra-option policy and termination probability. Because $\mu$ is deterministic, the corresponding row of the SMDP kernel is simply the option kernel:

$$e_{\bar{s}_0}^\top P_M^\mu = p_M^o(\,\cdot\,|\bar{s}_0), \qquad e_{\bar{s}_0}^\top P_{\widehat{M}}^\mu = p_{\widehat{M}}^o(\,\cdot\,|\bar{s}_0).$$

We now bound $\|p_M^o(\,\cdot\,|\bar{s}_0) - p_{\widehat{M}}^o(\,\cdot\,|\bar{s}_0)\|_1$ under Assumptions 4.3 and 4.4. Under Assumption 4.4, by repeated application of the triangle inequality,

$$d_{\mathcal{Z}}(Z_{t+j}, z_0) \leqslant j\sigma \quad \text{for all } j \geqslant 0.$$

For a fixed horizon $k \geqslant 1$, let $P_{s,\text{true}}^{(k)}(\cdot \mid \bar{s}_0, o)$ and $P_{s,\text{freeze}}^{(k)}(\cdot \mid \bar{s}_0, o)$ denote the $k$–step distributions of the primitive state $S_{t+k}$ when executing $o$ from $\bar{s}_0$ under $M$ (true drifting inputs) and under $\widehat{M}$ (frozen input), respectively.

Condition on a realization of the input path $(Z_t, Z_{t+1}, \ldots, Z_{t+k})$ in the true model and let $z_j := Z_{t+j}$. Under the conditioning on $z_{0:k}$, the state dynamics over the $k$ steps can be written as a product of inhomogeneous Markov kernels $K_j$ and $\widehat{K}_j$, $j = 0, \ldots, k-1$, where

$$K_j(s, \cdot) := P_s(\cdot \mid s, \pi_o(s, z_j), z_j),$$
$$\widehat{K}_j(s, \cdot) := P_s(\cdot \mid s, \pi_o(s, z_0), z_0).$$

Then

$$P_{s,\text{true}}^{(k)}(\cdot \mid \bar{s}_0, o) = \delta_{s_0} K_0 K_1 \cdots K_{k-1}, \qquad P_{s,\text{freeze}}^{(k)}(\cdot \mid \bar{s}_0, o) = \delta_{s_0} \widehat{K}_0 \widehat{K}_1 \cdots \widehat{K}_{k-1},$$

with $\delta_{s_0}$ the Dirac distribution at $s_0$. A standard telescoping argument using submultiplicativity for products of Markov kernels yields

$$\left\|\delta_{s_0} K_0 K_1 \cdots K_{k-1} - \delta_{s_0} \widehat{K}_0 \widehat{K}_1 \cdots \widehat{K}_{k-1}\right\|_1 \leqslant \sum_{j=0}^{k-1} \|K_j - \widehat{K}_j\|_\infty,$$

where again, $\|A\|_\infty = \max_i \sum_j |A_{ij}|$. By Assumption 4.3 and the drift bound $d_{\mathcal{Z}}(z_j, z_0) \leqslant j\sigma$,

$$\|K_j - \widehat{K}_j\|_\infty = \max_s \left\|P_s(\cdot \mid s, \pi_o(s, z_j), z_j) - P_s(\cdot \mid s, \pi_o(s, z_0), z_0)\right\|_1 \leqslant L_{\text{dyn}}\, d_{\mathcal{Z}}(z_j, z_0) \leqslant L_{\text{dyn}}\, \sigma j.$$

Thus, pathwise,

$$\left\|P_{s,\text{true}}^{(k)}(\cdot \mid \bar{s}_0, o) - P_{s,\text{freeze}}^{(k)}(\cdot \mid \bar{s}_0, o)\right\|_1 \leqslant L_{\text{dyn}}\, \sigma \sum_{j=0}^{k-1} j = L_{\text{dyn}}\, \sigma\, \frac{k(k-1)}{2}.$$

Since this bound is uniform in the input-path realization, it also holds after averaging over input paths.

Now consider the option-level discounted kernels. Under $M$ and $\widehat{M}$, for any $\bar{s}' = (s', z')$,

$$p_M^o(\bar{s}'|\bar{s}_0) = \sum_{k=1}^\infty \beta_o(1 - \beta_o)^{k-1}\gamma^k\, \mathbb{P}_M(\bar{S}_{t+k} = \bar{s}' \mid \bar{S}_t = \bar{s}_0, O_t = o),$$

$$p_{\widehat{M}}^o(\bar{s}'|\bar{s}_0) = \sum_{k=1}^\infty \beta_o(1 - \beta_o)^{k-1}\gamma^k\, \mathbb{P}_{\widehat{M}}(\bar{S}_{t+k} = \bar{s}' \mid \bar{S}_t = \bar{s}_0, O_t = o).$$

Applying the triangle inequality and exchanging sums,

$$\left\| p_M^o(\,\cdot\mid \bar{s}_0) - p_{\widehat{M}}^o(\,\cdot\mid \bar{s}_0)\right\|_1 = \sum_{z'}\sum_{s'}\left| p_M^o(\bar{s}_0,(s',z')) - p_{\widehat{M}}^o(\bar{s}_0,(s',z'))\right|$$

$$\leqslant \sum_{z'}\sum_{s'}\sum_{k=1}^{\infty}\beta_o(1-\beta_o)^{k-1}\gamma^k\left|\begin{array}{l}\mathbb{P}_M(\bar{S}_{t+k}=(s',z')\mid \bar{S}_t=\bar{s}_0, O_t=o)\\ \quad -\mathbb{P}_{\widehat{M}}(\bar{S}_{t+k}=(s',z')\mid \bar{S}_t=\bar{s}_0, O_t=o)\end{array}\right|$$

$$= \sum_{k=1}^{\infty}\beta_o(1-\beta_o)^{k-1}\gamma^k\sum_{z'}\sum_{s'}\left|\begin{array}{l}\mathbb{P}_M(\bar{S}_{t+k}=(s',z')\mid \bar{S}_t=\bar{s}_0, O_t=o)\\ \quad -\mathbb{P}_{\widehat{M}}(\bar{S}_{t+k}=(s',z')\mid \bar{S}_t=\bar{s}_0, O_t=o)\end{array}\right|.$$

Fix $k \geqslant 1$ and consider the joint distribution of the augmented termination state $\bar{S}_{t+k} = (S_{t+k}, Z_{t+k})$. Let $z_{0:k} = (z_0, z_1, \ldots, z_k)$ denote an input path of length $k$ starting at $z_0$, and write

$$w(z_{1:k}\mid z_0) := \prod_{j=0}^{k-1} P_z(z_{j+1}\mid z_j).$$

Under the true model $M$, conditioning on $z_{0:k}$ yields the state distribution $P_{s,\mathrm{true}}^{(k)}(\cdot\mid \bar{s}_0, o)$ as above, and the terminal input is $Z_{t+k} = z_k$. Thus, for any $(s', z')$,

$$\mathbb{P}_M(S_{t+k}=s', Z_{t+k}=z'\mid \bar{S}_t=\bar{s}_0, O_t=o) = \sum_{\substack{z_{1:k-1}\\ z_k=z'}} w(z_{1:k}\mid z_0)\, P_{s,\mathrm{true}}^{(k)}(s'\mid \bar{s}_0, o;\, z_{0:k}),$$

where we emphasize that $P_{s,\mathrm{true}}^{(k)}(\cdot\mid \bar{s}_0, o;\, z_{0:k})$ denotes the $k$–step state distribution conditional on the particular input-path realization $z_{0:k}$. Under $\widehat{M}$, by Definition 4.2, we use the same input-path law for the terminal input (equivalently, the same weights $w(z_{1:k}\mid z_0)$), but evolve the state using the frozen input $z_0$. Hence,

$$\mathbb{P}_{\widehat{M}}(S_{t+k}=s', Z_{t+k}=z'\mid \bar{S}_t=\bar{s}_0, O_t=o) = \sum_{\substack{z_{1:k-1}\\ z_k=z'}} w(z_{1:k}\mid z_0)\, P_{s,\mathrm{freeze}}^{(k)}(s'\mid \bar{s}_0, o),$$

since $P_{s,\mathrm{freeze}}^{(k)}(\cdot\mid \bar{s}_0, o)$ does not depend on the intermediate inputs $z_{1:k}$.

Therefore,

$$\sum_{z'}\sum_{s'}\left|\mathbb{P}_M(S_{t+k}=s', Z_{t+k}=z'\mid \bar{S}_t=\bar{s}_0, O_t=o) - \mathbb{P}_{\widehat{M}}(S_{t+k}=s', Z_{t+k}=z'\mid \bar{S}_t=\bar{s}_0, O_t=o)\right|$$

$$= \sum_{z'}\sum_{s'}\left|\sum_{\substack{z_{1:k-1}\\ z_k=z'}} w(z_{1:k}\mid z_0)\left(P_{s,\mathrm{true}}^{(k)}(s'\mid \bar{s}_0, o;\, z_{0:k}) - P_{s,\mathrm{freeze}}^{(k)}(s'\mid \bar{s}_0, o)\right)\right|$$

$$\leqslant \sum_{z'}\sum_{s'}\sum_{\substack{z_{1:k-1}\\ z_k=z'}} w(z_{1:k}\mid z_0)\left|P_{s,\mathrm{true}}^{(k)}(s'\mid \bar{s}_0, o;\, z_{0:k}) - P_{s,\mathrm{freeze}}^{(k)}(s'\mid \bar{s}_0, o)\right| \qquad \text{(triangle inequality)}$$

$$= \sum_{z_{1:k}} w(z_{1:k}\mid z_0)\sum_{s'}\left|P_{s,\mathrm{true}}^{(k)}(s'\mid \bar{s}_0, o;\, z_{0:k}) - P_{s,\mathrm{freeze}}^{(k)}(s'\mid \bar{s}_0, o)\right|$$

$$= \sum_{z_{1:k}} w(z_{1:k}\mid z_0)\left\|P_{s,\mathrm{true}}^{(k)}(\cdot\mid \bar{s}_0, o;\, z_{0:k}) - P_{s,\mathrm{freeze}}^{(k)}(\cdot\mid \bar{s}_0, o)\right\|_1$$

$$\leqslant L_{\mathrm{dyn}}\,\sigma\,\frac{k(k-1)}{2},$$

where the last inequality uses the uniform pathwise bound established above.

Plugging this back into the option-level bound yields

$$\left\|p_M^o(\,\cdot\,|\bar{s}_0) - p_{\widehat{M}}^o(\,\cdot\,|\bar{s}_0)\right\|_1 \leqslant \sum_{k=1}^{\infty} \beta_o(1-\beta_o)^{k-1}\gamma^k\, L_{\mathrm{dyn}}\,\sigma\,\frac{k(k-1)}{2}$$

$$= L_{\mathrm{dyn}}\,\sigma \sum_{k=1}^{\infty} \beta_o(1-\beta_o)^{k-1}\gamma^k\,\frac{k(k-1)}{2}.$$

Let $r := \gamma(1-\beta_o)$. Then

$$\beta_o(1-\beta_o)^{k-1}\gamma^k = \beta_o\gamma r^{k-1},$$

and the remaining series is

$$\sum_{k=1}^{\infty} \beta_o(1-\beta_o)^{k-1}\gamma^k\,\frac{k(k-1)}{2} = \frac{\beta_o\gamma}{2}\sum_{k=1}^{\infty} k(k-1)r^{k-1}.$$

Using the identity $\sum_{k=1}^{\infty} k(k-1)r^{k-1} = \frac{2r}{(1-r)^3}$ for $|r| < 1$, we obtain

$$\sum_{k=1}^{\infty} \beta_o(1-\beta_o)^{k-1}\gamma^k\,\frac{k(k-1)}{2} = \beta_o\gamma\frac{r}{(1-r)^3} = \frac{\beta_o\gamma^2(1-\beta_o)}{\left(1-\gamma(1-\beta_o)\right)^3}.$$

Therefore

$$\left\|p_M^o(\,\cdot\,|\bar{s}_0) - p_{\widehat{M}}^o(\,\cdot\,|\bar{s}_0)\right\|_1 \leqslant L_{\mathrm{dyn}}\,\sigma\,\frac{\beta_o\gamma^2(1-\beta_o)}{\left(1-\gamma(1-\beta_o)\right)^3}.$$

To obtain a simpler bound, note that

$$1 - \gamma(1-\beta_o) = (1-\gamma) + \gamma\beta_o \geqslant \beta_o,$$

so

$$\frac{\beta_o\gamma^2(1-\beta_o)}{\left(1-\gamma(1-\beta_o)\right)^3} \leqslant \frac{\beta_o\gamma^2}{(\beta_o)^3} = \frac{\gamma^2}{\beta_o^2}.$$

Since $\beta_o \geqslant \beta_{\mathrm{eval}}$, we have $\beta_o^2 \geqslant \beta_{\mathrm{eval}}^2$ and hence

$$\frac{1}{\beta_o^2} \leqslant \frac{1}{\beta_{\mathrm{eval}}^2}.$$

Thus, for every $\bar{s}_0$,

$$\left\|e_{\bar{s}_0}^\top P_M^\mu - e_{\bar{s}_0}^\top P_{\widehat{M}}^\mu\right\|_1 = \left\|p_M^o(\,\cdot\,|\bar{s}_0) - p_{\widehat{M}}^o(\,\cdot\,|\bar{s}_0)\right\|_1 \leqslant \frac{L_{\mathrm{dyn}}\gamma^2\sigma}{\beta_{\mathrm{eval}}^2}.$$

Taking the maximum over $\bar{s}_0$ yields

$$\|P_M^\mu - P_{\widehat{M}}^\mu\|_\infty = \max_{\bar{s}_0}\left\|e_{\bar{s}_0}^\top(P_M^\mu - P_{\widehat{M}}^\mu)\right\|_1 \leqslant \frac{L_{\mathrm{dyn}}\gamma^2\sigma}{\beta_{\mathrm{eval}}^2}.$$

Combining this bound with (10), we obtain

$$\max_\mu \|V_{M,\mathcal{O}_{\mathrm{eval}}}^\mu - V_{\widehat{M},\mathcal{O}_{\mathrm{eval}}}^\mu\|_\infty \leqslant \frac{R_{\max}}{(1-\gamma)^2} \cdot \frac{L_{\mathrm{dyn}}\gamma^2\sigma}{\beta_{\mathrm{eval}}^2}.$$

Finally, recalling the factor 2 from the reduction to optimal meta-policies at the beginning of the proof, we conclude that

$$\left\|V_{M,\mathcal{O}_{\mathrm{eval}}}^{\mu_{M,\mathcal{O}_{\mathrm{eval}}}^\star} - V_{M,\mathcal{O}_{\mathrm{eval}}}^{\mu_{\widehat{M},\mathcal{O}_{\mathrm{eval}}}^\star}\right\|_\infty \leqslant \frac{2R_{\max}L_{\mathrm{dyn}}\gamma^2\sigma}{(1-\gamma)^2\beta_{\mathrm{eval}}^2}.$$

This establishes the claimed variance bound under the frozen-input approximation with per-step input drift.

# G. Proof of Theorem 4.6

Recall Equation 2:

$$
\underbrace{\left\| V_{M,\mathcal{O}}^{\mu_{M,\mathcal{O}}^{\star}} - V_{M,\mathcal{O}}^{\mu_{\widehat{M}}^{\star},\mathcal{O}_{\text{eval}}} \right\|_{\infty}}_{\text{commitment loss}} \leqslant \underbrace{\left\| V_{M,\mathcal{O}}^{\mu_{M,\mathcal{O}}^{\star}} - V_{M,\mathcal{O}}^{\mu_{M,\mathcal{O}_{\text{eval}}}^{\star}} \right\|_{\infty}}_{\text{deliberation cost}} + \underbrace{\left\| V_{M,\mathcal{O}_{\text{eval}}}^{\mu_{M,\mathcal{O}_{\text{eval}}}^{\star}} - V_{M,\mathcal{O}_{\text{eval}}}^{\mu_{\widehat{M}}^{\star},\mathcal{O}_{\text{eval}}} \right\|_{\infty}}_{\text{variance}}
\tag{11}
$$

$$\tag{12}$$

We use Lemma 2.5 and Lemma 4.5 to get:

$$
\underbrace{\left\| V_{M,\mathcal{O}}^{\mu_{M,\mathcal{O}}^{\star}} - V_{M,\mathcal{O}}^{\mu_{\widehat{M}}^{\star},\mathcal{O}_{\text{eval}}} \right\|_{\infty}}_{\text{commitment loss}} \leqslant \underbrace{\left\| V_{M,\mathcal{O}}^{\mu_{M,\mathcal{O}}^{\star}} - V_{M,\mathcal{O}}^{\mu_{M,\mathcal{O}_{\text{eval}}}^{\star}} \right\|_{\infty}}_{\text{deliberation cost}} + \underbrace{\left\| V_{M,\mathcal{O}_{\text{eval}}}^{\mu_{M,\mathcal{O}_{\text{eval}}}^{\star}} - V_{M,\mathcal{O}_{\text{eval}}}^{\mu_{\widehat{M}}^{\star},\mathcal{O}_{\text{eval}}} \right\|_{\infty}}_{\text{variance}}
\tag{13}
$$

$$
\leqslant \frac{\gamma C_{\max}}{1-\gamma} \left( \beta_{\text{eval}} - \beta_{\min} \right) + \frac{2 R_{\max} L_{\text{dyn}} \gamma \sigma}{(1-\gamma)^2 \beta_{\text{eval}}^2}
\tag{14}
$$

# H. Proof of Corollary 4.7

By Theorem 4.6, for any $\beta_{\text{eval}} \in [\beta_{\min}, 1]$ we have

$$
\left\| V_{M,\mathcal{O}}^{\mu_{M,\mathcal{O}}^{\star}} - V_{M,\mathcal{O}}^{\mu_{\widehat{M}}^{\star},\mathcal{O}_{\text{eval}}} \right\|_{\infty} \leqslant \frac{\gamma C_{\max}}{1-\gamma} \left( \beta_{\text{eval}} - \beta_{\min} \right) + \frac{2 R_{\max} L_{\text{dyn}} \gamma^2 \sigma}{(1-\gamma)^2 \beta_{\text{eval}}^2}
$$

$$
= -A\beta_{\min} + A\beta_{\text{eval}} + \frac{B}{\beta_{\text{eval}}^2},
$$

where

$$
A := \frac{\gamma C_{\max}}{1-\gamma}, \qquad B := \frac{2 R_{\max} L_{\text{dyn}} \gamma^2 \sigma}{(1-\gamma)^2}
$$

The term $-A\beta_{\min}$ does not depend on $\beta_{\text{eval}}$, so minimizing the bound over $\beta_{\text{eval}} \in [\beta_{\min}, 1]$ is equivalent to minimizing

$$
g(\beta) := A\beta + \frac{B}{\beta^2}, \qquad \beta \in [\beta_{\min}, 1].
$$

Since $A > 0$ and $B > 0$, $g$ is differentiable on $(0, \infty)$ with

$$
g'(\beta) = A - \frac{2B}{\beta^3}, \qquad g''(\beta) = \frac{6B}{\beta^4} > 0 \quad \text{for all } \beta > 0.
$$

Thus $g$ is strictly convex on $(0, \infty)$ and has at most one critical point. Solving $g'(\beta) = 0$ yields

$$
A - \frac{2B}{\beta^3} = 0 \quad \Longrightarrow \quad \beta^3 = \frac{2B}{A} \quad \Longrightarrow \quad \beta_0 := \left( \frac{B}{A} \right)^{\frac{1}{3}}.
$$

Because $g''(\beta_0) > 0$, this critical point is the unique global minimizer of $g$ on $(0, \infty)$.

We now restrict to the feasible interval $[\beta_{\min}, 1]$. There are three cases:

1. If $\beta_0 \in [\beta_{\min}, 1]$, then by convexity the minimum of $g$ on $[\beta_{\min}, 1]$ is attained at $\beta_0$.

2. If $\beta_0 < \beta_{\min}$, then $g'(\beta) > 0$ for all $\beta \geqslant \beta_{\min}$, so $g$ is strictly increasing on $[\beta_{\min}, 1]$ and the minimum is attained at $\beta_{\min}$.

3. If $\beta_0 > 1$, then $g'(\beta) < 0$ for all $\beta \leqslant 1$, so $g$ is strictly decreasing on $[\beta_{\min}, 1]$ and the minimum is attained at 1.

In all cases, the minimizer over $[\beta_{\min}, 1]$ is given by

$$\beta_{\text{eval}}^{\star} = \text{clip}_{[\beta_{\min},1]}(\beta_0) = \text{clip}_{[\beta_{\min},1]}\left(\frac{B}{A}\right)^{\frac{1}{3}} .,$$

where we define the clipping operator

$$\text{clip}_{[\beta_{\min},1]}(x) := \min\{1, \max\{\beta_{\min}, x\}\}.$$

we obtain, for fixed $(\gamma, |\mathcal{S}|, |\mathcal{A}|, \delta)$,

$$\beta_{\text{eval}}^{\star} = \Theta\left(\left(\frac{R_{\max}L_{\text{dyn}}\sigma}{C_{\max}}\right)^{\frac{1}{3}}\right).$$

# I. Hyperparameters for CatAndMouse experiment

| Parameter | Value |
|---|---|
| *Environment* | |
| Grid size | $10 \times 10$ |
| Step reward ($r_{\text{step}}$) | $-0.2$ |
| Catch reward ($r_{\text{catch}}$) | $10.0$ |
| Max episode steps | $50$ |
| *Options* | |
| Number of options ($|\Pi|$) | $8$ |
| Termination probabilities ($B$) | $\{0.1, 0.2, \ldots, 1.0\}$ |
| *Network Architecture* | |
| Position embedding dim | $16$ |
| Direction embedding dim | $4$ |
| Hidden layers | $2$ |
| Hidden layer size | $64$ |
| Activation function | ReLU |
| Output size | $80 \ (8 \times 10)$ |
| *Training* | |
| Algorithm | Q-learning |
| Optimizer | Adam |
| Learning rate ($\alpha$) | $0.001$ |
| Discount factor ($\gamma$) | $0.99$ |
| Exploration ($\varepsilon$) | $0.1$ |
| Episodes | $150,000$ |
| Seeds | $5$ |

