# OpenReview forum: "The Cost of Commitment in Option-Based Hierarchical RL"
_ICML.cc/2026/Conference — ICML 2026 regular_

### Official Review · Reviewer_cKDD · 2026-02-21

**Soundness:** 2
**Presentation:** 3
**Significance:** 2
**Originality:** 3
**Overall Recommendation:** 4
**Confidence:** 4

**Summary:**

This paper studies the tradeoff between deliberation cost and compounding model error in option-based hierarchical reinforcement learning (HRL). The authors assume the setting in which the termination probability of options is constant over states, which is relevant for action-repeating options. The authors define commitment loss as the value gap between (i) an optimal meta-policy learned with the full option class under true dynamics and (ii) an optimal meta-policy learned with restricted option durations under an approximate model, but evaluated in the true environment. The paper introduces novel PAC-style bounds on the commitment loss when the model is learned from finite samples, as well as a complementary analysis of such bounds for Input-driven SMDP (IDSMDP). Experiments are performed in tabular MDPs to further validate the theoretical results.

**Compliance With Llm Reviewing Policy:**

Affirmed.

**Final Justification:**

Due to the author's clarifications during the rebuttal, I have increased my score to 4. I am not confident enough to further raise my score since assuming state-independent termination functions may limit the interest of the HRL community in the paper.

**Key Questions For Authors:**

- The paper assumes that using a model is a common approach in option-based RL, but most works are model-free. I suggest motivating the relevance of considering a model-based approach to improve the flow of the paper.

- Although assumption 2.1 is satisfied by options constructed from action repetitions, one can argue that this is a specific niche among the options literature. The main motivation for options is the decomposition of complex behavior into different, simpler skills that can be chained together. The paper would be significantly strengthened if the authors could present partial results for $\beta(s)$ rather than a constant $\beta$.

- “The intuition behind Definition 4.2 is to model scenarios where inputs are assumed constant between option boundaries, while costly observation at termination collapses uncertainty to the true input distribution.” This sentence is hard to understand. Please consider rewriting it for improved clarity.

- Given the simplicity of the domains, why use only 5 different seeds to report the results?

**Limitations:**

yes

**Strengths And Weaknesses:**

### Strengths
- The paper is clearly written and organized.
- To the best of my knowledge, the theoretical results are novel and sound. Mainly, separating the commitment loss into a deliberation cost term (linear in $\beta$) and a variance term (inverse in $\beta$) makes the tradeoff for option termination probabilities transparent.
- The resulting U-shaped curves and optimal $\beta^*$ are intuitive and analytically tractable. This particular result may be of interest to the HRL community.

### Weaknesses

- The work relies on a few strong assumptions, the most relevant being the assumption that the termination function $\beta$ is constant over states. While the authors acknowledge such limitations, they limit the applicability and remove the flexibility that makes options attractive. For instance, the results are not applicable to settings in which options terminate after reaching some particular goal state or achieving some particular task.
- The PAC bounds depend on worst-case ℓ₁ perturbation results and scale with $1/\beta$ or $1/\beta^2$. It is unclear how tight these are in practice. The theory predicts qualitative trends, but quantitative guidance for real systems may be weak.
- Experiments were conducted in toy MDPs (e.g., 10x10 grid). It would strengthen the paper to include larger tabular environments or continuous control settings to assess whether the reported trends scale.

---

> ### Author Rebuttal · Authors · 2026-03-30
>
> Thank you for your feedback on our paper.
>
>
> Answer to first key question: This is a fair critique. The reason we have opted for the model-based setting is because it directly allows us to mathematically model how errors compound with the commitment length. In the model-free case, the transition model is still there, but it is implicit (see this ICML 2025 paper [1]). The model-free setting therefore makes it more difficult to reason directly about the tradeoff. Additionally, in the Cat-And-Mouse experiment, we use a model-free DQN, and our scaling laws are still qualitatively holding. However, we agree that a better indication of why we chose the model-based framework for the theory would explicitly help the manuscript. As such, we propose to add more background about this in the introduction.
>
> The second key question is in most part answered by our response to reviewer RqDZ (key question 1). The core argument is that to achieve a bound that can take into account arbitrary forms of terminations, you would need to account for the coupling between the policy and the duration of the option. This leads to the specific structure of the MDP mattering, and making it very difficult to talk about different option lengths. Although this is probably possible by looking at the properties of one specific domain (e.g. robotics), as a first bound tackling this problem in the literature, we wanted to provide general scaling laws that are domain agnostic. This follows precedent in the MDP literature [2,3,4], which begin with strong structural assumptions (tabular or linear MDP, generative model access, ergodicity) when opening a new analytical direction, and subsequent work relaxing these assumptions over time. We see our contribution in the same light.
>
> Third key question: This is indeed a bit verbose. We propose the following reformulation: “The intuition behind Definition 4.2 is to model scenarios where uncertainty increases with the duration of options, while paying the cost of option termination collapses uncertainty to the true input distribution”.
>
> Fourth key question: We observed very low variance across seeds (as visible in the tight confidence intervals). To make sure, we reran the experiment with 10 seeds (std = 0.0463), and the curves look almost identical. The monotonic relationships are still holding, confirming the robustness of the result.
>
> [1] Richens, J., Abel, D., Bellot, A., & Everitt, T. (2025). General agents contain world models. arXiv preprint arXiv:2506.01622.
>
> [2] Jiang, N., Singh, S., & Tewari, A. (2016). On Structural Properties of MDPs that Bound Loss Due to Shallow Planning. In Proceedings of IJCAI.
>
> [3] Jiang, N., Kulesza, A., Singh, S., & Lewis, R. (2015). The dependence of effective planning horizon on model accuracy. In Proceedings of AAMAS.
>
> [4] Hu, H., Yang, Y., Zhao, Q., & Zhang, C. (2022). On the role of discount factor in offline reinforcement learning. In Proceedings of ICML.

---

> > ### Author Rebuttal · Reviewer_cKDD · 2026-04-02
> >
> > I thank the authors for addressing some of my questions.
> >
> > I understand that tackling arbitrary forms of termination functions would be very challenging. Although the bounds are domain agnostic, they are not agnostic to the type of options used; on the contrary, the provided bounds are restricted only to action-repeating options. I believe this should be stated more clearly in the paper. For instance, if one reads the title and the abstract of this paper, one might think that the results are valid for general types of options.
> >
> > Could you please elaborate more on the significance of action-repeating options, or how the theoretical results could be useful and further extended in future work?

---

> > > ### Author Response · Authors · 2026-04-02
> > >
> > > We would like to thank you for the questions, these are very useful.
> > >
> > >
> > > > Although the bounds are domain agnostic, they are not agnostic to the type of options used; on the contrary, the provided bounds are restricted only to action-repeating options.
> > >
> > > We want to clarify that Assumption 2.1 restricts the termination function, not the intra-option policies. The termination function considered in the action-repetition setting [1] is one example satisfying this assumption, but our Definition 2.2 ($\mathcal{O} := \Pi \times B$) admits any finite set of intra-option policies $\Pi$, including complex state-dependent policies (navigation goal, locomotion policy, etc). The restriction is that termination probabilities are state-independent, which is strictly weaker than requiring action-repeating options which also require a restriction on the policy class (playing the same action no matter the state).
> > > > Could you please elaborate more on the significance of action-repeating options, or how the theoretical results could be useful and further extended in future work?
> > > # Significance:
> > > State-independent terminations arise naturally beyond the action-repetition setting; it can be found in any setting where the commitment length is a design choice decoupled from the skill itself. In LLM-based agents, plan-and-execute architecture [2] commit to a multi-step plan before re-invoking the LLM, where the commitment length trades off token and latency cost against plan staleness. In robotics, it is also standard practice to replan at a fixed frequency dictated by computational budget, for instance running SLAM or MPC at a rate decoupled from the low-level control loop. Although we do not claim to fully answer the question regarding a complex domain like LLMs or robotics, Section 4 characterizes the deliberation cost vs input drift tradeoff which recalls the tradeoffs faced in these domains. In all of these settings, our scaling laws (Corollaries 3.3 and 4.7) give practitioners a principled way to reason about the tradeoff and tune the replanning frequency depending on the environment.
> > >
> > > # Extensions:
> > > As discussed in our response to the acknowledgement of reviewer ijWD, we believe the commitment loss decomposition (Equation 2) could serve as a foundation for an extension to state-dependent termination, with the deliberation cost requiring additional structure from the MDP and policy class. We also note that our Cat-and-Mouse experiment (Figure 2) already operates in a regime with neural function approximation and the qualitative prediction of our theory holds cleanly, which we find encouraging for the generality of the scaling laws beyond the strict assumptions of the current proofs.
> > >
> > > We could strengthen these points explicitly in our paper if you believe that it would clarify questions for future readers.
> > >
> > > [1] Biedenkapp, A., Rajan, R., Hutter, F., and Lindauer, M. (2021) Temporl: Learning when to act. In Proceedings of ICML.
> > >
> > > [2] Xu, B., Peng, Z., Lei, B., Mukherjee, S., Liu, Y., & Xu, D. (2023). Rewoo: Decoupling reasoning from observations for efficient augmented language models. arXiv preprint arXiv:2305.18323.

---

### Official Review · Reviewer_ijWB · 2026-02-24

**Soundness:** 4
**Presentation:** 3
**Significance:** 2
**Originality:** 3
**Overall Recommendation:** 4
**Confidence:** 3

**Summary:**

This paper introduces a theoretical framework to formalize the tradeoff between paying greater deliberation costs with shorter options, and making suboptimal decisions due to persisting with options that may no longer be optimal. The authors decompose the "Commitment Loss" (loss in value due to deliberation cost and model error) into two components: loss due to greater deliberation cost from shorter options given a perfect model, and the loss due to model error given shorter options. Bounds on these two components are presented, and termination rate thresholds that minimize the upper bound on the total commitment loss in model-based RL and IDMDP settings. Two experiments are presented to support the theoretical results.

**Compliance With Llm Reviewing Policy:**

Affirmed.

**Final Justification:**

The rebuttal reinforced my prior assessment. I recommend accept because I think that novel and correct theoretical results are worth publishing. I remain at weak accept and confidence 3 because I am concerned about the limited scope, and the authors have provided some not entirely satisfactory responses to this. The "more complex environment" that the theory can predict is still very toy, and the suggestions about how the theory could be extended are not detailed or concrete enough to give me more confidence about it.

**Key Questions For Authors:**

Can the authors say anything about how tight the upper bound on the commitment loss is?

**Limitations:**

the authors do not discuss many limitations mentioned in the first bullet of the Weaknesses section

**Strengths And Weaknesses:**

Strengths:
- The paper provides novel theoretical results about a foundational question in hierarchical RL: The trade-off between deliberation cost and model error as a function of option duration.
- Theoretical results are backed up by both proofs and experiments, while also explaining trends observed in prior work
- The writing is clear, with intuitive explanations for the implications of the theory

Weaknesses:
- The applicability and generality of the results are very limited:
    - to discrete state and action space settings.
    - assuming options are chosen by planning through an approximate dynamics model, but many approaches using options do not do explicit model-based planning, e.g., [1]
    - The assumption of state-independent termination is very limiting, ruling out many natural types of option e.g. options for reaching bottleneck states. This is even more limiting considering the additional assumption that all options can be initiated from any state. This rules out options that have any kind of precondition, because they cannot be equally likely to succeed when initiated from states that satisfy their preconditions as states that do not.

- The decomposition into deliberation cost and variance seems a bit arbitrary. Why not instead define variance as the value loss due to model error with the full set of options, and the deliberation cost as the value loss due shorter options given the imperfect model? As currently defined, the deliberation cost assumes the model is perfect, which is not realistic.

- Half the results are specific to the setting of Input-Driven MDPs; these should be defined in the Background section rather than introduced for the first time in Section 4.

[1] Bacon, Pierre-Luc, Jean Harb, and Doina Precup. "The option-critic architecture." Proceedings of the AAAI conference on artificial intelligence. Vol. 31. No. 1. 2017.

---

> ### Author Rebuttal · Authors · 2026-03-29
>
> Answer to the key question: We do not have a formal tightness result, and we want to be transparent about that. However, we believe this is consistent with the contribution the paper is making. The goal of our bounds is not to provide tight numerical estimates of the commitment loss (like one could see in the PAC-Bayes literature), but rather characterize the structure of the tradeoff: How it scales with option durations, the discount factor, the structure of costs versus rewards, the way inputs change, etc. The scaling laws (Corollaries 3.3 and 4.7) we derive from the bounds are the main deliverable, and the experiments confirm that the qualitative behavior matches observations in practice.
>
> This is analogous to how bounds in the planning and approximation literature are typically used (see [1,2]). The value of Theorems 3.2 and 4.6 (commitment loss from model error and input drift, respectively) is in identifying which quantities matter and how they interact. Tightening the bounds and adapting it to domain specific niches is a natural follow up work.
>
> [1 ] Hu, H., Yang, Y., Zhao, Q., & Zhang, C. (2022). On the role of discount factor in offline reinforcement learning. In Proceedings of ICML.
>
> [2] Gheshlaghi Azar, M., Munos, R., & Kappen, H. J. (2013). Minimax PAC bounds on the sample complexity of reinforcement learning with a generative model. Machine learning, 91(3), 325-349

---

> > ### Author Rebuttal · Reviewer_ijWB · 2026-04-01
> >
> > Thanks for answering my question. I have two questions remaining, relating to points I brought up in the Weaknesses section:
> >
> > 1. The current definition of the deliberation cost assumes the model is perfect, which is unrealistic - what are the implications of this assumption? Why not instead define variance as the value loss due to model error with the full set of options, and the deliberation cost as the value loss due shorter options given the imperfect model?
> >
> > 2. How hard would it be to build on this work to extend to common practical HRL settings? You mention in your response to reviewer RqDZ that extending toward state-dependent termination would require a completely different set of assumptions. In that case, to what extent could this meaningfully build on your work?

---

> > > ### Author Response · Authors · 2026-04-01
> > >
> > > We would like to thank you for your questions. We really appreciate it.
> > >
> > > >The current definition of the deliberation cost assumes the model is perfect, which is unrealistic - what are the implications of this assumption?
> > >
> > > We want to be clear that there are actually no assumptions made on the form of the deliberation cost. The quantity we want to upper bound is the Commitment Loss (Definition 2.4). To get there we introduce and remove the term $V_{M,\mathcal{O}}^{\mu^*_{M,\mathcal{O}_{\text{eval}}}}$, which naturally splits into two quantities: one we call the deliberation cost, the other one we call the variance (see Equation 2). In short, we have defined the commitment loss, in a way that naturally recovers a term we call the deliberation cost, and this term essentially targets the cost from short commitments on a perfect model. This is very similar to previous work in the RL literature, where one term measures the error from using a low discount factor on the perfect model, and the other measures the variance from using a high discount factor on an imperfect environment (see [1] theorem 2, and [2] theorem 3.3)
> > >
> > > > Why not instead define variance as the value loss due to model error with the full set of options, and the deliberation cost as the value loss due shorter options given the imperfect model?
> > >
> > > We could possibly recover the commitment loss with this alternative decomposition, as previous work [1,2,3] on the traditional bias-variance also admits several decompositions of the same quantity (typically the planning loss). We chose the proposed decomposition to highlight that the commitment loss can be seen as a tradeoff between the cost to pay for changing options frequently (which we called the deliberation cost) and the potential loss caused by staying in options for a long time (which we called the variance). We found this decomposition on the perfect model more intuitive as it strictly separates the deliberation cost and model error, isolating  the impact of option duration on both terms.
> > >
> > > > How hard would it be to build on this work to extend to common practical HRL settings? You mention in your response to reviewer RqDZ that extending toward state-dependent termination would require a completely different set of assumptions. In that case, to what extent could this meaningfully build on your work?
> > >
> > > This is a great question. We empirically have provided support that the current form of the scaling laws (Corollary 3.3, Corollary 4.7) still qualitatively hold in simple (Figure 1.) and more complex environments (Figure 2.). This means that today, practitioners can start wondering about the shape of rewards versus deliberation cost, $L_{\text{dyn}}, \sigma$ and the various other quantities in their environment. Our results are already useful today to tune termination probabilities and overall understand the interconnection with deliberation costs, providing a deeper understanding of the deliberation cost, which was introduced [4] (and is still commonly used) as a heuristic. We want to be clear that providing a first theoretical understanding of the impact of options duration on the tradeoff between the deliberation cost and the uncertainty reduction  was our main goal for this paper.
> > >
> > > That being said, our work could serve as a basis for an extension to state-dependent terminations. We believe that the decomposition of the commitment loss could be reused, albeit with a deliberation cost term that would depend on the policy class and some structural properties of the MDP to effectively control their impact on the option durations. The variance terms (for both section 3 and 4) could remain somewhat unchanged, as they already consider the worst case (longest options possible in the set).
> > >
> > > [1] Jiang, N., Kulesza, A., Singh, S., & Lewis, R. (2015). The dependence of effective planning horizon on model accuracy. In Proceedings of AAMAS.
> > >
> > > [2] Hu, H., Yang, Y., Zhao, Q., & Zhang, C. (2022). On the role of discount factor in offline reinforcement learning. In Proceedings of ICML.
> > >
> > > [3] Lefebvre, R., & Durand, A. (2025). On shallow planning under partial observability. In Proceedings of AAAI
> > >
> > > [4] Harb, J., Bacon, P. L., Klissarov, M., & Precup, D. (2018). When waiting is not an option: Learning options with a deliberation cost. In Proceedings of AAAI

---

### Official Review · Reviewer_RqDZ · 2026-03-12

**Soundness:** 2
**Presentation:** 2
**Significance:** 2
**Originality:** 3
**Overall Recommendation:** 3
**Confidence:** 3

**Summary:**

This paper studies how long an agent should stay with an option when switching options has a cost, but using long options can also lead to model error or outdated information. The paper introduces a “commitment loss” and breaks it into two parts: a deliberation-cost term and a variance term. It then provides bounds for two settings: finite-sample model error in an MDP, and input drift in an input-driven SMDP using a frozen-input approximation. The paper also derives closed-form scaling laws for the termination threshold and presents experiments in a Ring MDP and a Cat-and-Mouse environment that qualitatively match the theory.

**Compliance With Llm Reviewing Policy:**

Affirmed.

**Final Justification:**

The rebuttal clarified the paper’s intended scope and claim. However, my main concerns remain about the narrow setting studied and the limited empirical support, so I keep my initial recommendation.

**Key Questions For Authors:**

* How essential is Assumption 2.1? Could any part of the main result still hold even in weaker form with state-dependent termination?

* Why is Cat-and-Mouse the right validation for the input-drift theorem? Could you add a simpler controlled experiment where
$L_{dyn}$ or drift can be measured directly, so the theorem can be tested more directly?

**Limitations:**

yes

**Strengths And Weaknesses:**

The main idea is clean and easy to follow. The two settings are also reasonable: one focuses on statistical model error, and the other on stale external inputs. The final scaling laws are intuitive: when deliberation cost is higher, the analysis favors longer options; when error or drift is larger, it favors shorter options.

The analysis relies on strong assumptions. The most important is state-independent termination, where options are defined only by a constant termination probability and fixed intra-option policies that are reused across durations. This is a fairly narrow class of options, and it is unclear how much of the conclusion would still hold for standard learned options with state-dependent termination. The drift result depends on Lipschitz closed-loop dynamics together with a per-step almost sure drift bound. These assumptions are mathematically convenient, but restrictive. So the theory appears sound only in a limited regime, and the paper should be more explicit about this gap.

I was also not fully convinced by the experimental validation. The Ring experiment is small and synthetic. The Cat-and-Mouse experiment uses a DQN-style learner and reports average optimal termination probabilities over states, but the connection between this practical setup and the exact theory is loose. There is also no comparison with stronger HRL baselines that adapt option length, and no study of how tight the bounds are. The appendix includes hyperparameters, but overall the empirical section feels somewhat light given the scope of the claims.

---

> ### Author Rebuttal · Authors · 2026-03-29
>
> Thank you for your feedback, we are happy you found the paper easy to follow and liked the scaling laws we recovered.
>
> Answer to key question 1:  This is a question we ourselves have thought about when writing up the paper. The short answer is no, the current results do not extend to state-dependent termination in their present form, and this is by design rather than oversight.
> The core difficulty is that state-dependent termination couples option durations with the option’s policy (because visiting certain states can trigger early termination). This also means that the duration distribution becomes entangled with the specific state structure of the environment, and any bound would need to account for that coupling explicitly. The resulting theorem would no longer be domain agnostic, it would depend tightly on the properties of the particular MDP.
>
> To the best of our knowledge, this is the first paper to formally characterize the deliberation cost vs model/drift error tradeoff for option duration. Therefore, we deliberately chose assumptions that yield clean, general scaling laws. This follows precedent in the MDP literature we take inspiration from [1,2,3]. These works similarly begin with strong structural assumptions (tabular or linear MDP, generative model access, ergodicity) when opening a new analytical direction, with subsequent work relaxing these over time. We see our contribution in the same light.
>
> We have some early ideas for extending toward state-dependent termination, but it would require a completely different set of assumptions, not a straightforward generalization. We do acknowledge your point, and will clarify this boundary more explicitly by adding the following sentences to Section 2.1:
>
> [OLD]  “Options constructed from action repetitions (Biedenkapp et al., 2021) typically satisfy this assumption. Given state-independent terminations, it is natural to talk about the duration of an option, which is on the order of $\frac{1}{\beta(o)}$”
>
> [NEW] “... Options constructed from action repetitions (Biedenkapp et al., 2021) typically satisfy this assumption. Although quite restrictive, removing this assumption leads to a tight coupling between the option’s duration, the policy and the state structure. In this work, we focus on a domain agnostic theoretical understanding of how option duration affects deliberation costs and compounding errors. We leave the exploration of domain specific bounds for future work.
> Given state-independent terminations, it is natural to talk about the duration of an option, which is on the order of $\frac{1}{\beta(o)}$ …”
>
> Answer to key question 2: This is a fair point. We want to clarify what the cat-and-mouse experiment is and isn’t intended to support.
> The goal of that experiment is not to verify the bound quantitatively, but to test whether the qualitative prediction of Corollary 4.7 (that optimal termination rate would increase with the input variability) transfers to a regime far outside the assumptions of Theorem 4.6 (neural network approximation, high-dimensional states, etc). The fact that this monotonic relationship holds cleanly in that setting is, we would argue, stronger evidence for the practical relevance of the scaling laws that we uncovered than a tighter verification in a toy domain, where the assumptions would hold by construction.
> While measuring $L_{dyn}$ directly is technically possible, the value of the experiment lies precisely in using environment-level proxies (mouse direction that can change at different rates) as heuristics rather than computing exact bound quantities. We argue this is the regime a practitioner would actually be in. However, we acknowledge that a complementary controlled experiment measuring drift would be a more direct support to the theorem.
>
> [1] Jiang, N., Singh, S., & Tewari, A. (2016). On Structural Properties of MDPs that Bound Loss Due to Shallow Planning. In Proceedings of IJCAI.
>
> [2] Jiang, N., Kulesza, A., Singh, S., & Lewis, R. (2015). The dependence of effective planning horizon on model accuracy. In Proceedings of AAMAS.
>
> [3] Hu, H., Yang, Y., Zhao, Q., & Zhang, C. (2022). On the role of discount factor in offline reinforcement learning. In Proceedings of ICML.

---

> > ### Author Rebuttal · Reviewer_RqDZ · 2026-04-03
> >
> > Thank you for the response. My remaining concerns are mainly about scope and empirical support. The paper still studies a narrow option class with state-independent termination. While your response explains this choice, I remain concerned that this limits relevance to more standard HRL settings. The empirical support still feels somewhat limited, ie the Ring domain is very small, and the Cat-and-Mouse result only shows a qualitative trend, not direct evidence that the theory is practically predictive. I would most like to see a controlled experiment testing whether the termination rate predicted by the scaling law is close to the empirically optimal rate, since this would directly validate the paper’s main theoretical claim rather than only its qualitative trend.

---

> > > ### Author Response · Authors · 2026-04-04
> > >
> > > Thank you for your questions.
> > >
> > > > While your response explains this choice, I remain concerned that this limits relevance to more standard HRL settings
> > >
> > > In the current form, our work models any situations where skills/options are decoupled from their duration. We detail some examples in our response to cKDD, but to summarize, it applies in cases where you make a plan with a pre-decided length, and execute it. We believe this is a relevant class of problems even if it doesn’t encompass all of HRL so far.
> > >
> > > > The empirical support still feels somewhat limited, i.e. the Ring domain is very small, and the Cat-and-Mouse result only shows a qualitative trend, not direct evidence that the theory is practically predictive. I would most like to see a controlled experiment testing whether the termination rate predicted by the scaling law is close to the empirically optimal rate, since this would directly validate the paper’s main theoretical claim rather than only its qualitative trend.
> > >
> > > As detailed in our answer to ijWB, we want to be transparent that the goal of this work is not to provide tight numerical estimates of the commitment loss, but rather to characterize the structure of the tradeoff. We wanted to explain how the commitment loss scales with option durations, the discount factor, the structure of cost versus rewards, the way inputs drift over time, etc. The scaling laws (Corollaries 3.3 and 4.7) we derive from the bounds are the main deliverable, and the experiments confirm that the qualitative behavior matches observations in practice.
> > >
> > > This is similar to prior theoretical work in the planning and approximation literature (see [1,2]). The value of Theorems 3.2 and 4.6 (commitment loss from model error and input drift, respectively) is in identifying which quantities matter and how they interact. We agree that work tightening the numerical estimates using more complex statistical tools could be valuable future work. However, for papers introducing a new framework like ours, it is not typical to look at predictive power. They typically want to see if the relationships between the different quantities still hold in practice, which is what our experiments do.
> > >
> > > [1 ] Hu, H., Yang, Y., Zhao, Q., & Zhang, C. (2022). On the role of discount factor in offline reinforcement learning. In Proceedings of ICML.
> > >
> > > [2] Jiang, N., Kulesza, A., Singh, S., & Lewis, R. (2015). The dependence of effective planning horizon on model accuracy. In Proceedings of AAMAS.

---

### Official Review · Reviewer_3GEc · 2026-03-12

**Soundness:** 3
**Presentation:** 2
**Significance:** 3
**Originality:** 3
**Overall Recommendation:** 4
**Confidence:** 3

**Summary:**

The authors address a not very well-explored problem in option-based HRL, where a long running option can be suboptimal and incur losses despite being optimal at initiation.  The paper analyses the tradeoff between the deliberation cost, i.e., the cost of having to terminate and re-plan and the model variance, which increases with the running time of options. They additionally introduce and formalise commitment loss and provide proofs showing the use of shorter options can reduce variance from both model error and drift error.

**Compliance With Llm Reviewing Policy:**

Affirmed.

**Key Questions For Authors:**

See Weakness 2.

**Limitations:**

yes

**Strengths And Weaknesses:**

Strengths: The primary contribution of the paper is the formalisation of the commitment loss, which helps in understanding the tradeoff while running options with long durations in the HRL setting, and provides PAC bound on this loss in terms of $\beta_{\text{eval}}$ (the termination rate parameter). The paper makes a novel contribution in understanding a gap in the literature, in this particular setting. The literature review is fairly thorough, and problem setting is well motivated, and the proofs provided seem sound. The experimental setup is well explained and authors also provide the relevant hyper parameters for reproducibility.


Weaknesses:
1) The experimental setup seems a bit limited.
2) As mentioned, Harutyunyan et al.[1] studies a similar setting, which also allows this flexibility in timescale for options and analyses the bias-variance tradeoffs. It would perhaps be useful if the technical contributions of this paper and key differences can be highlighted.
2) It would be a bit more readable to have the related literature in one place, preferably in the beginning.


[1] Harutyunyan, A., Vrancx, P., Hamel, P., Nowé, A., & Precup, D. (2019). Per-Decision Option Discounting. International Conference on Machine Learning.

---

> ### Author Rebuttal · Authors · 2026-03-29
>
> We would like to thank you for your feedback on our paper.
>
>
> Response to your key question (also identified weakness 2): Harutyunyan et al. (2019) introduce a new way of discounting in HRL. As the traditional primitive discount factor makes it hard to reason over long horizons (which HRL is supposed to help with!), they provide an alternative by decoupling discounting for rewards and transitions. We study a different question by focusing on the tradeoff between deliberation cost and model/drift error. However, both our work and Harutyunyan’s build on similar literature  (bias-variance from temporal discounting, discount pessimism, etc), which can be confusing. We propose to modify the following sentences in related works (Section 6) to clarify this:
>
>  [Old] ”Despite this huge interest and body of work in the MDP setting, the study of generalization performance in the context of HRL has received almost no attention other than Harutyunyan et al. (2019) which draws a link between option durations and the discount factor and gives bias and variance bounds.”
>
> [New] “Despite this huge interest and body of work in the MDP setting, the study of generalization performance in the context of HRL has received almost no attention other than Harutyunyan et al. (2019). They introduce a new discounting framework by decoupling discount factors for rewards and transitions within options, and derive bias-variance bounds on the resulting estimation error. Our work is complementary: rather than modifying the discounting scheme, we study how option durations interact with model or drift error, and derive bounds on the performance loss due to termination.”

---

> > ### Author Rebuttal · Reviewer_3GEc · 2026-04-04
> >
> > I thank the authors for the answer. I maintain my score for acceptance.

---

> > > ### Author Response · Authors · 2026-04-05
> > >
> > > We thank the reviewer again for the positive assessment.

---

### Decision · Program_Chairs · 2026-04-30

**Decision:**

Accept (regular)

**Comment:**

The paper studies the cost of commitment in hierarchical RL and provides the first formalization of trade-off between deliberation cost and model drift in option-based RL. The reviewers have generally appreciated the theoretical contributions of the paper and the results are sound. Although one reviewer raised the concern that the benchmarks used were not large-scale, I think they highlight the main concepts quite well. The resulting U-shaped curves are quite intuitive and easy to follow. Therefore, despite the lack of large-scale experimental evaluation, I recommend accepting the work.